# Genomic surveillance uncovers regional variation in HCV transmission networks in rural United States

Damien C. Tully [1,2] ✉, David J. Bean[3], Jacklyn Sarette[3], Thang Long Ngo[3], Karen A. Power[3], Daniel Brook[4], Hannah Cooper[5], Judith Feinberg[6], Peter D. Friedmann [7], Karli R. Hochstatter[8], Jennifer R. Havens[9], Shanna Babalonis[9], Christopher Hurt[10], Wiley Jenkins[11], P. Todd Korthuis[12], William Miller [13], Mai T. Pho[14], Gordon Smith [15], Thomas J. Stopka[16], Judith I. Tsui[17], L. Sarah Mixson[17], Ryan P. Westergaard[18], April M. Young[19] & Todd M. Allen[3]

Hepatitis C virus (HCV) remains a public health concern in the United States, particularly in rural communities where the opioid epidemic accelerates transmission among people who use drugs (PWUD). Despite this growing burden, the genetic features and transmission dynamics of HCV in these settings are poorly understood. We analyze 692 HCV antibody-positive specimens collected from rural communities in ten U.S. states using amplicon-based deep sequencing and the Global Hepatitis Outbreak and Surveillance Technology (GHOST) platform to reconstruct transmission networks. Among sequenced individuals, 29.5% are linked within clusters. Cluster structure varies by region from sparse networks in Ohio to dense clusters in New England and phylogenetic analyses show that some networks persist for over a decade, indicating sustained transmission. Nearly half of all clusters involve individuals connected through social recruitment, suggesting peer-referral strategies effectively identify transmission chains. Penalized regression retains only a few individual factors including younger age, peer or partner recruitment, illegal income, methamphetamine use, each with modest effects. These findings suggest that clustering is shaped primarily by social and structural contexts rather than individual characteristics and underscore the importance of integrating genomic surveillance with social-network insights to detect emerging HCV clusters and guide targeted interventions in underserved rural communities.

Hepatitis C presents a significant global health challenge, affecting an estimated 58 million individuals who are chronically infected with the hepatitis C virus (HCV)[1]. Notably, out of the estimated 11 million people worldwide who engage in injecting drug use each year, nearly 40% have viremic HCV infection[2]. The transmission of HCV is greatly influenced by unsafe injecting practices among persons who use drugs (PWUD), making it a substantial contributor to new infections on a global scale[3]. Within the United States, HCV is the leading cause of liver cancer and death from liver disease. The incidence rate of acute hepatitis C in the US has doubled since 2014 (129% increase) and increased 7% from 2020 to 2021[4]. Rates of acute hepatitis C are highest among males, persons aged 20–39 years and those living in the Eastern

---

**Fig. 1 | Sampling overview of HCV from study sites across the United States.**
**A** Geographical representation of study sampling sites (OR Oregon, WI Wisconsin, IL Illinois, KY Kentucky, WV West Virgina, OH Ohio, NC North Carolina, VT Vermont, NH New Hampshire, MA Massachusetts). The map was generated using the Map-Chart (https://www.mapchart.net) and is licensed under a Creative Commons Attribution-ShareAlike 4.0 International License (https://creativecommons.org/licenses/by-sa/4.0/). **B** Sampling dates of blood specimen collection for different states. *X*-axis represents the sampling date and the *y*-axis lists the sampling locations. The New England study site is composed of Vermont, New Hampshire, and Massachusetts.

and Southeastern states[5]. Central to this increase is the increased frequency of reported injection drug use concurrent with the rise of prescription opioids and increasing availability of heroin/fentanyl and methamphetamine[6].

Rural areas bear a disproportionate burden of HCV, where infection rates are estimated to be twice as high as in urban settings[7]. These outbreaks often occur in communities facing structural barriers such as lower education and income levels, as well as limited access to healthcare. Additionally, rural populations in the US frequently lack essential harm reduction services like sterile syringe services programs and medication for opioid use disorder and often experience housing instability, which may exacerbate vulnerability to drug-related harms[8]. Recent studies reveal a significant prevalence of HCV among young adult PWUD in rural areas, particularly those engaging in polysubstance injection[9,10]. Recent outbreaks among persons who inject drugs in Scott County, Indiana[11], Lowell, Massachusetts[12] and Kanawha County, West Virginia[13] highlight a concerning trend with high rates of HCV infections often preceding HIV outbreaks among persons who inject drugs. This temporal association underscores the urgent need for proactive measures and interventions in rural communities to prevent the emergence of HIV outbreaks.

In response to the opioid crisis affecting rural areas of the US, a collaborative effort involving several agencies, including the National Institute on Drug Abuse (NIDA), the Centers for Disease Control and Prevention (CDC), the Substance Abuse and Mental Health Administration (SAMHSA), and the Appalachian Regional Commission (ARC), led to the establishment of the Rural Opioid Initiative (ROI). This initiative aimed to collect and synthesize both quantitative and qualitative data from eight rural regions across ten states, with the goal of deepening our understanding of drug use and the local factors driving opioid consumption, as outlined by Jenkins et al.[14].

As part of this initiative blood specimens were collected for the purpose of conducting rapid HIV, HCV, and syphilis testing. HCV-positive specimens were subsequently analyzed at a laboratory funded by the ROI for next-generation sequencing and linkage analysis to identify genetically associated transmissions. Drawing a parallel with the pandemic of COVID-19, genomic surveillance has been crucial in tracking the global spread of SARS-CoV-2, with real-time analysis forming a cornerstone for public health decision-making. However, similar genomic surveillance strategies, including genetic-based inferences, have not been routinely employed to investigate HCV transmission, except in specific community-based PWUD cohorts in

Baltimore[15,16], San Francisco[17], and correctional facilities in Wisconsin[18]. Due to the scarcity of both genomic and epidemiological data, our knowledge about the origins of these outbreaks, the dynamics of virus transmission, and the genetic diversity of HCV strains in rural US communities remains limited. The primary aim of this study was to characterize HCV strains circulating among PWUD in rural US areas and identify factors associated with HCV transmission clusters. To date, there are no studies surveying the genetic landscape of HCV across rural communities adversely affected by the US opioid crisis and whether the transmission patterns differ across geographic locations.

## Results

### Study participants

A total of 3048 PWUD completed audio computer-assisted self-interviews (ACASIs) or computer-assisted self-interviews (CASIs). Participants had a mean age of 34 years [IQR: 28–43]) and 42% of respondents were female. From each study site, a sub-sample of respondents were selected to give a plasma sample for further HCV sequencing and cluster analysis. A total of 1201 HCV-positive serum specimens were received from eight study sites (Fig. 1). Of these, 692 (57.7%) successfully completed sequencing and quality control, while 293 (24.4%) samples were found to be below our limit of detection and contained little if any viral RNA. The specimens may have represented HCV antibody-positive status but viral load-negative samples from those who had cleared infection. From 1201 specimens, 216 (18.0%) either failed PCR or did not generate sufficient sequence to pass GHOST QC for inclusion. The PCR-only failure rate was 11.4% overall, varying by site (3.37–46.15%; Supplementary Table 1), largely reflecting pre-analytical differences in specimen handling and RNA integrity.

Participant characteristics among those with available HCV sequencing and completed questionnaires (n = 429) are shown in Table 1. Overall, the median age was 35 years [IQR: 29–43] and 64% were male and mostly non-Hispanic white with a high school education. Sixty-one percent had experienced homelessness in the past 6-months and most participants were recruited into the study by a friend, associate, or acquaintance. No significant differences were observed between the sequenced samples and those that failed PCR amplification. We compared demographic characteristics across three groups: participants who contributed sequenced samples, those who contributed samples that were not sequenced, and those who did not contribute a sample. As shown in Table 1, the groups were broadly

**Table 1 | Demographic characteristics of participants from the Rural Opioid Initiative (ROI)**

| | Overall ROI cohort | ROI participants with samples | ROI participants without samples | p-value[a] | ROI participants with sequenced samples | p-value overall |
|---|---|---|---|---|---|---|
| | N (%) | N (%) | N (%) | - | N (%) | - |
| Total | 3048 | 737 (24) | 2311 (76) | | 429 (14) | |
| Age, mean (SD)[b] | 36.1 (10.3) | 36.8 (10.1) | 35.9 (10.3) | 0.04 | 36.8 (9.9) | 0.7 |
| Gender | | | | 0.6 | | 0.007 |
| Male | 1737 (57) | 428 (58) | 1309 (57) | | 273 (64) | |
| Female | 1293 (42) | 306 (42) | 987 (43) | | 155 (36) | |
| Transgender/Gender minority | 18 (1) | 3 (<1) | 15 (1) | | 1 (<1) | |
| Race/Ethnicity[c] | | | | 0.002 | | 0.009 |
| Non-Hispanic White | 2527 (83) | 621 (84) | 1906 (82) | | 364 (85) | |
| Non-Hispanic Black | 94 (3) | 13 (2) | 81 (4) | | 8 (2) | |
| Non-Hispanic American Indian | 208 (7) | 43 (6) | 165 (7) | | 20 (5) | |
| Non-Hispanic Other/Unknown | 103 (3) | 19 (3) | 84 (4) | | 14 (3) | |
| Hispanic[d] | 116 (4) | 41 (6) | 75 (3) | | 23 (5) | |
| Education | | | | 0.2 | | 0.3 |
| Less than high school | 688 (23) | 180 (24) | 508 (22) | | 94 (22) | |
| High school or GED[e] | 1430 (47) | 349 (47) | 1081 (47) | | 212 (49) | |
| Some college or technical school | 856 (28) | 195 (26) | 661 (29) | | 114 (27) | |
| Bachelor's degree or above | 71 (2) | 12 (2) | 59 (3) | | 8 (2) | |
| Marital status | | | | 0.08 | | 0.09 |
| Single/Not married | 1,570 (52) | 388 (53) | 1182 (51) | | 229 (53) | |
| Married | 354 (12) | 101 (14) | 253 (11) | | 57 (13) | |
| Separated, Divorced, Widowed, Don't Know | 1054 (35) | 239 (33) | 815 (35) | | 139 (32) | |
| Homelessness, past 6 months[f] | 1612 (53) | 434 (59) | 1178 (51) | <0.001 | 260 (61) | 0.007 |
| Geographic region | | | | <0.001 | | <0.001 |
| Illinois | 173 (6) | 12 (2) | 161 (7) | | 0 (0) | |
| Kentucky | 338 (11) | 42 (6) | 296 (13) | | 12 (3) | |
| North Carolina | 350 (11) | 51 (7) | 299 (13) | | 41 (10) | |
| New England[g] | 589 (19) | 228 (31) | 361 (16) | | 128 (30) | |
| Ohio | 258 (8) | 105 (14) | 153 (7) | | 63 (15) | |
| Oregon | 174 (6) | 57 (8) | 117 (5) | | 42 (10) | |
| Wisconsin | 991 (33) | 242 (33) | 749 (32) | | 143 (33) | |
| West Virginia | 175 (6) | 0 (0) | 175 (8) | | 0 (0) | |
| Recruited – How[h] | | | | | | |
| Coupon or Code | 2315 (76) | 571 (77) | 166 (23) | 0.3 | 342 (80) | 0.1 |
| Told about, but didn't get a coupon/code | 569 (19) | 128 (17) | 441 (19) | 0.3 | 69 (16) | 0.3 |
| Event | 32 (1) | 3 (<1) | 29 (1) | 0.05 | 1 (<1) | 0.1 |
| Online | 17 (1) | 6 (1) | 11 (<1) | 0.3 | 4 (1) | 0.5 |
| Flyer or other advertising | 96 (3) | 21 (3) | 75 (3) | 0.6 | 12 (3) | 0.9 |
| Other | 117 (4) | 27 (4) | 90 (4) | 0.8 | 13 (3) | 0.6 |
| Recruited – by Whom[h] | | | | | | |
| Partner, spouse, boyfriend, girlfriend | 326 (11) | 89 (12) | 237 (10) | 0.2 | 56 (13) | 0.04 |
| Casual sex partner | 44 (1) | 16 (2) | 28 (1) | 0.07 | 8 (2) | 0.02 |
| Friend, associate, acquaintance | 1706 (56) | 414 (56) | 1292 (56) | 0.7 | 242 (56) | 0.09 |
| Family member | 255 (8) | 53 (7) | 202 (9) | 0.1 | 30 (7) | 0.05 |
| Neighbor | 64 (2) | 12 (2) | 52 (2) | 0.3 | 6 (1) | 0.06 |
| Person I use drugs with | 398 (13) | 97 (13) | 301 (13) | 0.9 | 59 (14) | 0.09 |
| Service or program staff | 70 (2) | 10 (1) | 60 (3) | 0.04 | 9 (2) | 0.01 |
| Stranger | 37 (1) | 10 (1) | 27 (1) | 0.7 | 4 (1) | 0.05 |
| Other | 52 (2) | 9 (1) | 43 (2) | 0.2 | 5 (1) | 0.06 |

Shown are the overall characteristics from the entire ROI cohort compared to those who contributed samples and were sent to the Global Hepatitis Outbreak Surveillance Technology (GHOST) laboratory for sequencing analysis.

Continuous variables were compared using two-sided t-tests or one-way ANOVA, as appropriate. Categorical variables were compared using two-sided $\chi^2$ tests. P values are reported in the table. No adjustments were made for multiple comparisons.

[a]Tests participants who contributed a sample to those who did not.

[b]SD standard deviation.

[c]Race/ethnicity are mutually exclusive categories.

[d]Hispanic includes everyone who is Hispanic. White, Black, American Indian, and Other/Unknown race include those who are White, Black, American Indian, or Other/Unknown race and not Hispanic.

[e]GED, General Education Development.

[f]Defined as living from place-to-place, "couch-surfing", on the street, in a car, park, abandoned building, squat, or shelter.

[g]New England study site is composed of Vermont, New Hampshire and Massachusetts.

[h]Does not add up to 100% because participants could select more than one option.

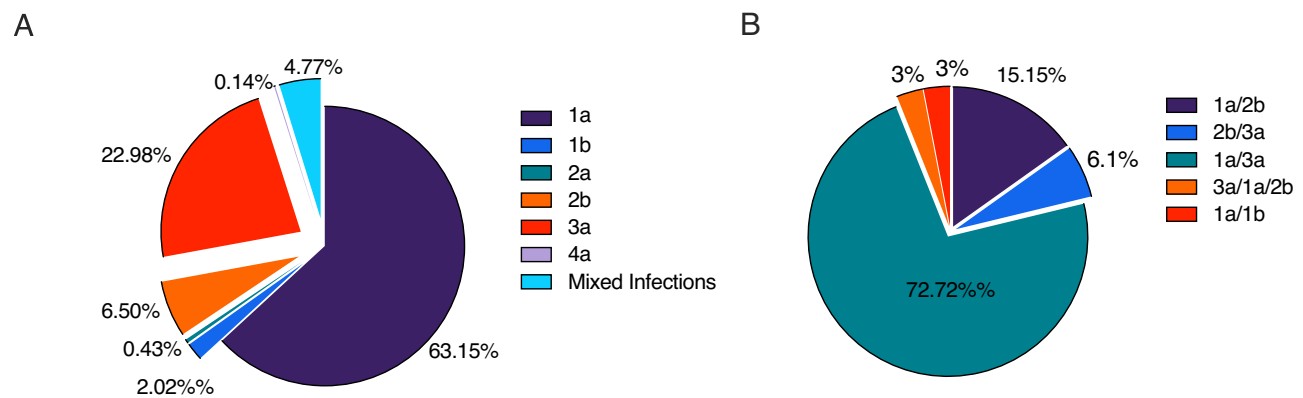

**Fig. 2 | Distribution of HCV genotypes across all study sites. A** Overall composition of genotypes (*n* = 692). **B** Breakdown of the 4.77% mixed infections by genotype (*n* = 33).

similar with respect to age and education. Statistically significant differences were observed for gender, race/ethnicity, homelessness, and geographic region. Specifically, males were slightly more likely to have sequenced samples compared to females, which likely reflects differences in sequencing yield and recruitment patterns across sites.

Most study participants reported using opioids (64%) followed by stimulants (33%) as their drugs of choice with a substantial proportion simultaneously injecting opioids and stimulants (Table 2). Median age at first injection drug use experience was 21 years [IQR: 18–32]. Eighty-two percent of participants reported accessing treatment for addiction, including inpatient/outpatient treatment and medication for opioid use disorder, with 77% reported health insurance or health care coverage. Most participants reported daily or more frequent injection drug use and received their syringes/needles from various sources including pharmacies, syringe services programs, friends/acquaintances, and drug dealers/street. Most participants were tested for HIV (77%) and HCV (75%). Only 3% were diagnosed with HIV, compared to 69% with HCV. Almost one-fifth (19%) cleared HCV infection with treatment.

### HCV genotype distribution
By utilizing Genome Detective tools for genotyping, we assigned consensus sequences to various HCV genotypes and subtypes as illustrated in Fig. 2. Out of the 692 sequences analyzed, 65% were classified as genotype 1, 23% as genotype 3, 7% as genotype 2 and one sequence was identified as genotype 4 (Fig. 2A). Our deep sequencing approach uncovered evidence of mixed infections in approximately 4.8% of cases (Fig. 2A). Further examination of these mixed infections revealed that the most prevalent combinations were genotype 1a and 3a, comprising 73% followed by genotype 1a/2b at 15% (Fig. 2B).

HCV genotype frequencies remained largely consistent among study sites with subtype 1a being the predominant strain (46–71%), followed by 3a (18–30%) and 2b (2–12%) (Supplementary Fig. 1). Although some variations in the prevalence of genotypes and subtypes were observed between sites, Kentucky and New England exhibited the highest variability. In Kentucky, we identified five different circulating genotypes, including a subtype 4a sample, and several low-prevalence mixed genotype infections (genotype 3a/1a, 2b/3a, 1a/2b) (Supplementary Fig. 1A). Similarly, New England displayed a range of circulating strains, with three subtype 2a samples and a higher frequency of subtype 2b samples compared to other sites. Despite the diversity of subtypes in circulation, only 2% of samples were found to be mixed infections (Supplementary Fig. 1B). Wisconsin showed relative homogeneity with only genotypes 1a, 3a and 2b detected while mixed infections were exclusively 3a/1a (Supplementary Fig. 1C). The remaining sites, Ohio, Oregon and North Carolina all exhibited a predominance of genotypes 1a and 3a with a similar frequency of genotype 2b followed by a range of mixed genotype infections (Supplementary Fig. 1D–F). In Oregon, a rare case of infection was found with a major population of genotype 3a and minor populations of genotypes 1a and 2b.

### Identification of recent transmission clusters
Phylogenetic analysis of consensus sequences indicates that study site specific sequences are interspersed throughout the tree although some sequences did appear to cluster by geographic location (Fig. 3). GHOST analysis of intra-host HCV HVR1 population from all sampled cases identified 85 transmission clusters involving 204 HCV strains (29.5%) (Fig. 4). The median cluster size was two members (range 2–7) with 63% in dyads. The fraction of clusters of HCV strains was statistically different according to study sites (*P* < 0.0001; Fisher's exact test) and ranged from sparse clustering in Ohio (9.4%) to almost half of all sequenced strains in New England (43%). Two study sites (Ohio and Oregon) exclusively consisted of dyads, while North Carolina had a single cluster of three. More complex networks were observed in Wisconsin, Kentucky, and New England, although they were limited. The largest cluster of PWUD was found in New England and consisted of a network of 7. Importantly, we believe that our targeted sequencing approach of HVR1 are most likely to capture recent transmission events as opposed to deeper historical connections.

### Comparison of identified clusters to RDS recruitment chains
Across all study sites, we found that 49% of transmission clusters were comprised of individuals linked within the RDS social recruitment chains. However, there was notable variability among the sampling sites in their ability to identify transmission networks using RDS recruitment. In Ohio and Oregon, where only dyadic genetic relationships were found, 50% and 67% of individuals, respectively, were not linked in the RDS chains. In Wisconsin, more complex networks were identified, with 60% found to be outside the RDS chains. Conversely, in New England and North Carolina, 55% and 77% of transmission clusters, respectively, were linked within the social recruitment chains. Overall, our observations indicated that there is a statistically significant (*P* = 0.0001) association between transmission cluster size and whether they are found inside or outside of the RDS chain. Specifically, individuals in more complex networks were more likely to be found in clusters that included members of their RDS chain compared to those in genetic dyads (70% vs. 36%, Supplementary Fig. 2).

### Relationship between intra-host viral diversity and clustering
The level of genomic diversity in HCV varies significantly depending on the infection stage. During the acute and early stages, the viral

**Table 2 | Substance use practices of sequenced Rural Opioid Initiative (ROI) participants**

| | ROI participants with sequenced samples |
|---|---|
| | N (%) |
| Total | 429 (58) |
| Substance Use | |
| Drug of choice | |
| Opioids | 274 (64) |
| Stimulant | 141 (33) |
| Benzodiazepines | 1 (<1) |
| Other | 13 (3) |
| Drug use patterns, past 30 days | |
| Opioids[a] | 370 (86) |
| Fentanyl | 197 (46) |
| Buprenorphine/methadone[b] | 225 (52) |
| Methamphetamine | 315 (73) |
| Cocaine/crack | 208 (48) |
| Benzodiazepines | 195 (45) |
| Multiple classes of drugs used[c] | 352 (82) |
| Number of classes, median (IQR)[d] | 3 (2–3) |
| Simultaneous injection of an opioid and a stimulant (e.g., speedball)[e,f] | 182 (45) |
| Personal history of overdose | 248 (58) |
| Ever received any treatment for addiction | 351 (82) |
| Received any treatment for addiction, past 30 days | 152 (35) |
| Attended inpatient/outpatient treatment, past 30 days | 109 (25) |
| Received medications for opioid use disorder, past 30 days | 99 (23) |
| Injection drug use | |
| Current injection drug use, past 30 days | 408 (95) |
| Injection drug use frequency, past 30 days[f] | |
| Daily or more | 297 (73) |
| More than weekly, less than daily | 40 (10) |
| Weekly | 26 (6) |
| Monthly | 39 (10) |
| Source of most syringes or needles, past 30 days[g] | |
| Pharmacy SSP/NEP[g], personally | 94 (23) |
| SSP/NEP[g], personally | 157 (38) |
| From someone else who got them from SSP/NEP[g] | 55 (13) |
| Friend/acquaintance, spouse, or partner | 27 (7) |
| Drug dealer/street | 67 (16) |
| Other | 4 (1) |
| Don't know/Refused | 4 (1) |
| Source of any syringes or needles, past 30 days[f] | |
| Pharmacy | 126 (31) |
| SSP/NEP[g], in person | 200 (49) |
| SSP/NEP[g], someone else | 140 (34) |
| Farm supply or veterinarian | 2 (<1) |
| Drug dealer or street syringe seller | 78 (19) |
| Spouse, partner, family member, or relative | 65 (16) |
| Friend or acquaintance | 143 (35) |
| I found them | 14 (3) |
| Person from whom you received most of your syringes was diabetic[f] | 15 (4) |
| Closest SSP[h] | |
| <30 min drive/Mobile SSP[h] comes to town | 313 (73) |
| >30 min drive | 65 (15) |
| Don't know | 37 (9) |
| Number of times got a new syringe from a pharmacy, past 30 days[h] | 0 (0–1) |

**Table 2 (continued) | Substance use practices of sequenced Rural Opioid Initiative (ROI) participants**

| | ROI participants with sequenced samples |
|---|---|
| Number of times got a new syringe from a SSP, past 30 days[h] | 0 (0–2) |
| It is easy for me to get new, clean syringes or needles | |
| Strongly/Somewhat disagree | 65 (15) |
| Uncertain | 42 (10) |
| Strongly/Somewhat agree | 319 (74) |
| Age at first injection, median (IQR)[h] | 21 (18–30) |
| Age at first opiate pain killer injection, median (IQR)[h] | 21 (17–28) |
| Age at first heroin injection, median (IQR)[h] | 24 (19–31) |
| Age at first methamphetamine injection, median (IQR)[h] | 26 (19–33) |
| Age at first cocaine injection, median (IQR)[h] | 22 (18–29) |
| Number of days practiced syringe mediated drug sharing, past 30 days[h], median (IQR) | 1 (0–8.5) |
| Number of days practiced multiple injection per injection episode, past 30 days[h], median (IQR) | 4 (1–15) |
| Number of times used syringe/needle that was used by somebody else, past 30 days[h], median (IQR) | 1 (0–5) |
| Number of times used a cotton, cooker, spoon, or water for rinsing or mixing that was used by somebody else, past 30 days[h], median (IQR) | 2 (0–10) |
| Number of times let someone else use a cotton, cooker, spoon, or water for rinsing or mixing after you used it, past 30 days[h], median (IQR) | 2 (0–10) |
| Main place received medical care, past 6 months | |
| Private doctor | 107 (25) |
| Community health center | 58 (14) |
| Health department | 19 (4) |
| Urgent care | 54 (13) |
| Emergency room | 83 (19) |
| Other | 27 (6) |
| Did not receive medical care in the past 6 months | 77 (18) |
| Refused/Don't know | 4 (1) |
| Health insurance or health care coverage | 330 (77) |
| Tested for HIV, ever | 331 (77) |
| Diagnosed with HIV, ever[i] | 11 (3) |
| Tested for HCV, ever | 332 (75) |
| Diagnosed with HCV, ever[j] | 222 (69) |
| Cleared HCV with treatment, ever[k] | 43 (19) |
| Rapid HCV test positive result | 424 (99) |
| Confirmatory RNA HCV positive result | 273 (64) |
| Rapid HIV positive results | 5 (1) |
| Confirmatory HIV positive results | 3 (1) |

[a]Heroin, opiate painkillers, and/or synthetics (e.g., U47700, U4, or "Pink").
[b]Buprenorphine and/or methadone used "to get high."
[c]Use of ≥2 drug categories (opioids, methamphetamine, cocaine/crack, prescription anxiety drugs [not as prescribed], gabapentin, clonidine, and/or other) by any route in past 30 days.
[d]IQR interquartile range.
[e]Simultaneous injection of an opioid and a stimulant (i.e., speedball, goofball, or screwball).
[f]Among participants reporting injection drug use in the past 30 days.
[g]SSP, Syringe Services Programs; NEP, Needle exchange programmes.
[h]Among participants reporting ever injecting drug. Overall ROI Cohort: n = 2812; ROI participants with a sample: n = 731; ROI participants with a sequenced sample: n = 426.
[i]Among participants reported ever being tested for HIV.
[j]Among participants reported ever being tested for HCV.
[k]Among participants reported ever being diagnosed with HCV.

population tends to be relatively homogeneous, primarily due to serial bottlenecks and the presence of a single founder virus. As the infection progresses, HCV undergoes increased genomic diversification as it adapts to the host's immune response, leading to a positive correlation between the stage of infection and intra-host viral diversity. Analysis of

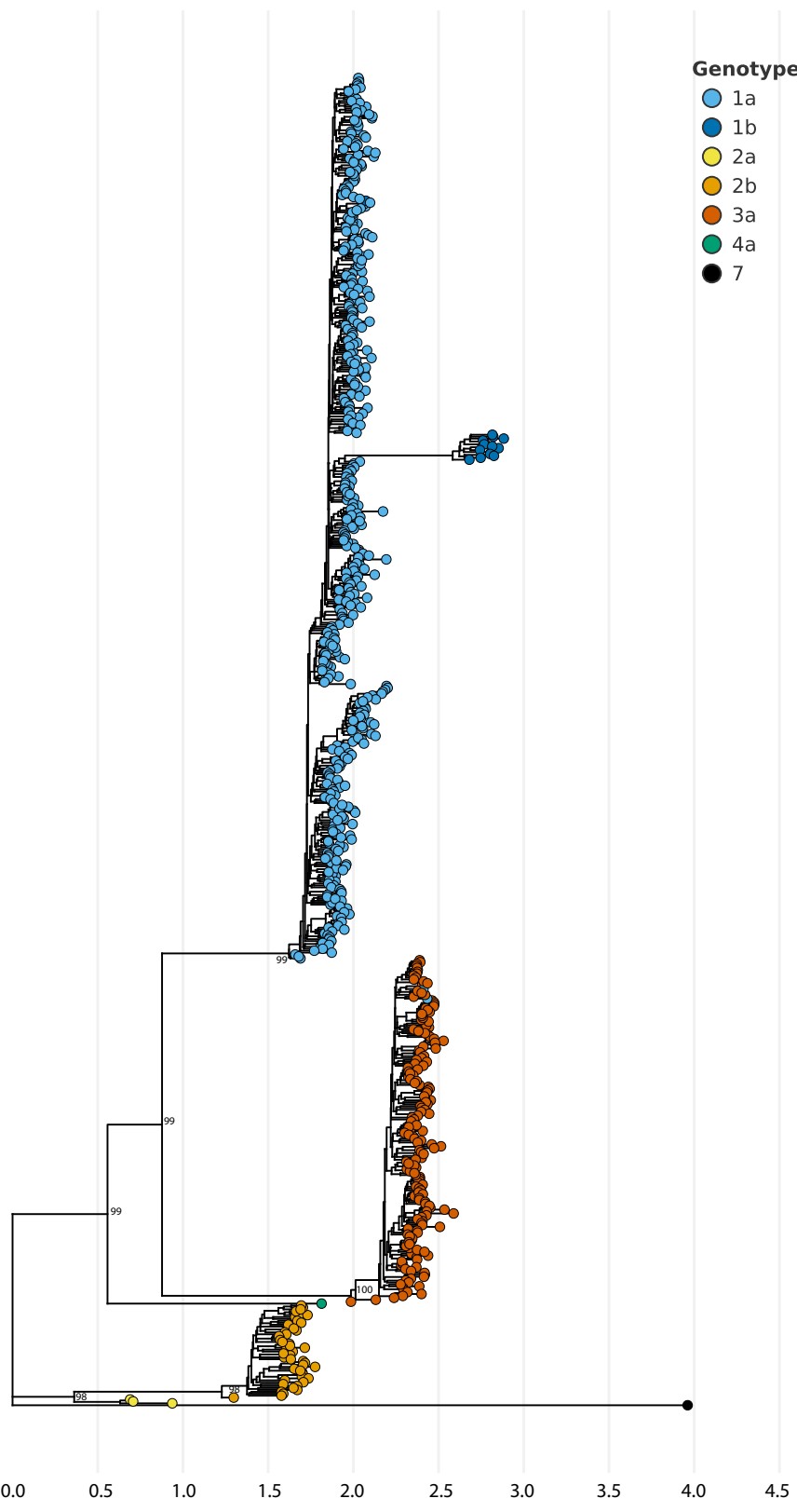

**Fig. 3 | Phylogenetic relationships between samples from individuals with HCV collected from the Rural Opioid Initiative, 2018–2021.** Maximum likelihood phylogenetic tree of consensus HCV sequences (306 bp) collected from the rural opioid initiative cohort. The tree was rooted on a genotype 7 reference sequence. Genotypes are labeled and shown by different colored tips. Only bootstrap support values greater than 70% are shown for major ancestral nodes. Scale bar indicates the number of substitutions per site.

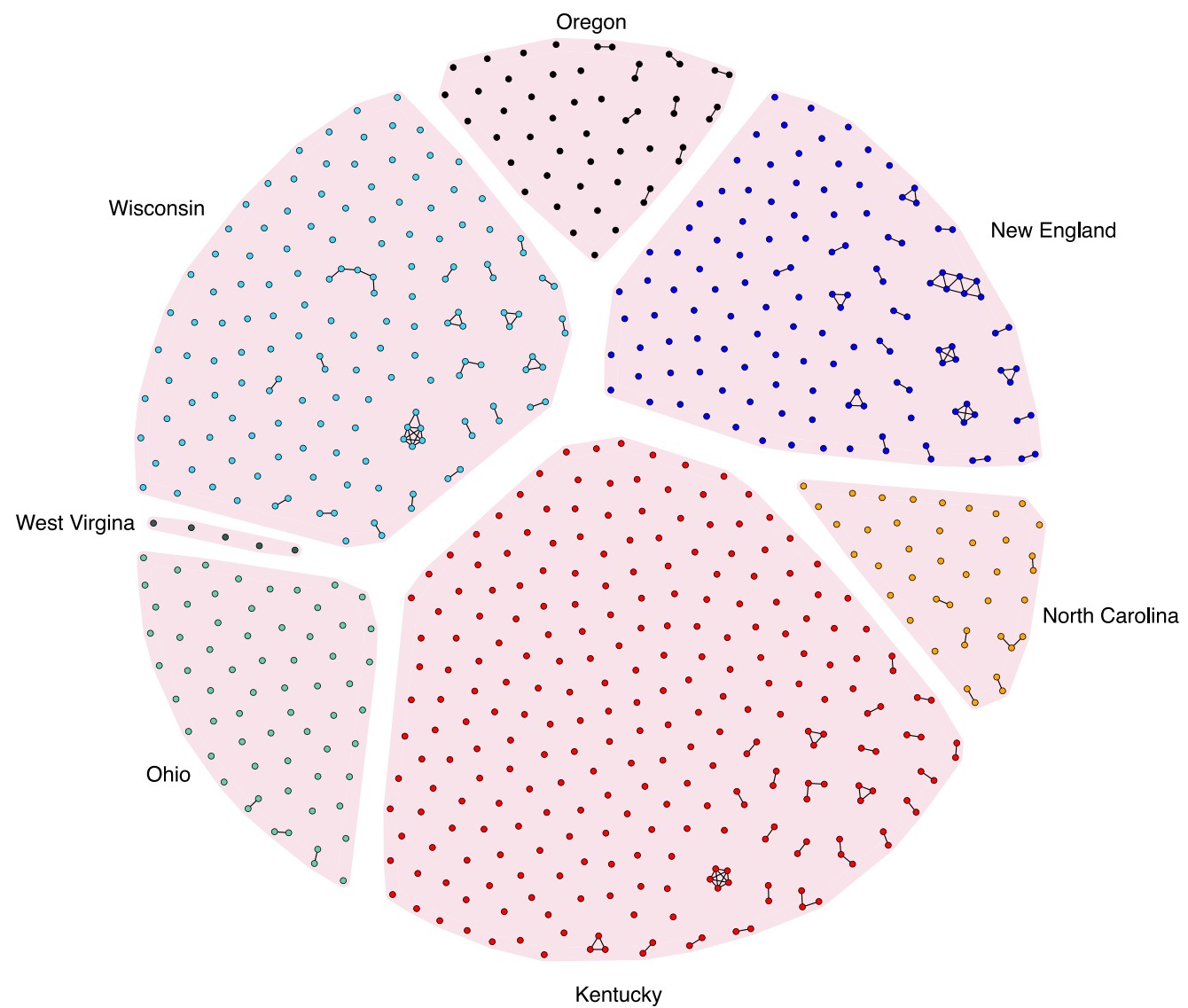

**Fig. 4 | HCV transmission networks as identified by Global Hepatitis Outbreak Surveillance Technology, 2018–2021.** Each node represents an HCV strain sampled from an individual study participant. A connecting line between nodes is drawn if the genetic distance between the strains is less than 0.037. Single unconnected dots represent samples for which no genetic linkage was detected within the specified threshold. Study sites are labeled, and color-coded as illustrated. The New England study site is composed of Vermont, New Hampshire, and Massachusetts.

intra-host viral diversity across all sampled individuals indicates that those who clustered exhibited significantly lower diversity compared to non-clustered individuals (Fig. 5A; $P = 0.0038$, Mann–Whitney test). Furthermore, the study revealed substantial variability in intra-host viral diversity among different study sites. Ohio and Kentucky (KY) showed the highest median diversity, whereas North Carolina displayed less heterogeneity. Despite a similar distribution of intra-host viral diversity between KY and New England, KY had a significantly higher median (Fig. 5B; $P = 0.0002$, Mann–Whitney test) a trend also observed when comparing KY with Wisconsin (WI) and NC (Fig. 5B). Further stratification of clustered individuals into dyads and more complex clusters (i.e., clusters with more than two members) demonstrated a significant difference in intra-host viral diversity between individuals not in a cluster and those in dyads (Fig. 5C; $P = 0.0161$, Mann–Whitney test) and a marginally non-significant difference for those in complex clusters ($P = 0.0506$, Mann–Whitney test). However, no significant difference was observed between dyads and more complex clusters ($P = 0.9269$, Mann–Whitney test). Taken together, this suggests that those participants not found to be in a transmission cluster may have an infection reminiscent of a longer timeframe (i.e., more chronic like stage of infection) compared to those within clusters who appear to be harboring less diversity which is a known attribute of the early stages of infection. However, this interpretation should be treated with caution as viral diversity can also be influenced by host immune pressure, treatment history, viral genotype, and potential technical biases introduced by amplicon sequencing. The observed site-specific differences in intra-host viral diversity likely reflect a combination of underlying epidemiological dynamics, transmission patterns and sampling variation rather than infection stage along.

## Evidence of HCV persistence across rural study sites

The rate of evolution for the genomic region used in this study was estimated to be $3.417 \times 10^{-3}$ (95% highest posterior density credibility intervals: $2.177$–$4.657 \times 10^{-3}$) substitutions per site per year. Using this rate, we estimated the tMRCA for each cluster and examined the lag time between inferred introduction date and time of first and last sampling date of the cluster (Fig. 6). The estimated tMRCA varied between clusters and sampling sites and the size of the cluster. Across all states, Oregon had the shortest median lag time of 3.61 years,

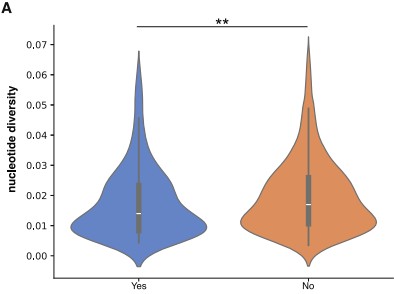
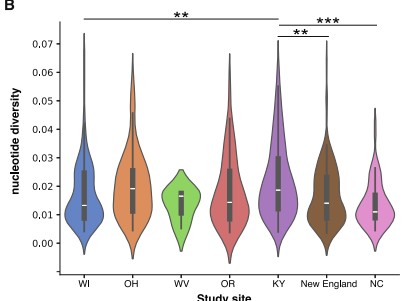
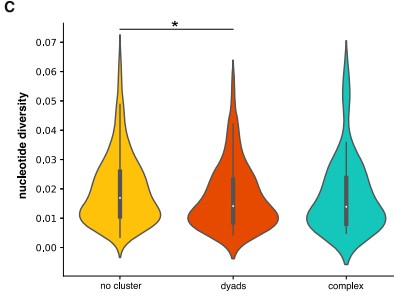

**Fig. 5 | Relationship between intra-host viral diversity and clustering illustrates that more recent infections may be driving transmission clusters.**
**A** Comparison of intra-host viral diversity between individuals who are part of a cluster and those who are not ($P = 0.0038$, $n = 692$). **B** Intra-host viral diversity across different study sites ($P = 0.0002$, $n = 691$) (WI Wisconsin, OH Ohio, WV West Virginia, OR Oregon, KY Kentucky, NC North Carolina). **C** Clustered individuals segregated into cluster size either dyads or more complex clusters (>2 individuals) ($P = 0.0161$, $n = 691$). Violin plots show the distribution of nucleotide diversity values by cluster type. The white line represents the median, the thick black bar indicates the interquartile range (IQR), and the thin black lines (whiskers) represent the data range excluding outliers. Statistical comparisons were performed using a two-sided Mann–Whitney U test (non-parametric) without adjustment for multiple comparisons. Exact $P$ values are shown above. Significance levels are indicated as *$P < 0.05$; **$P < 0.01$; ***$P < 0.001$; ***$P < 0.0001$. The New England study site is composed of Vermont, New Hampshire, and Massachusetts.

followed by Kentucky (4.15 years), North Carolina (4.80 years), New England (5.74 years), and Wisconsin (6.74 years), while the virus persisted longer in Ohio at 8.75 years.

In WI, one cluster had the longest persistence time of approximately 12.91 years while two clusters have the shortest lag times of less than a year (Fig. 6). In New England, a cluster comprising a dyad had the longest persistence time of 19.08 years while two dyad clusters had the shortest tMRCA until sampling of less than 6 months (Fig. 6). In North Carolina the tMRCA was estimated for four clusters with two having a tMRCA within 3 years while the remaining clusters had a persistence time of over 7 and 12 years respectively (Fig. 6). Within KY a cluster comprising 3 participants was found to be persisting for approximately 13 years (tMRCA mid-2005) before this cluster was sampled in 2019. In comparison, a dyad cluster was found to have the shortest lag time within the inferred introduction mirroring the time of sampling (Fig. 6). Three transmission clusters were detected in OH where only once was estimated to have a more recent introduction occurring an estimated 1.7 years prior to sampling (Fig. 6). The remaining two clusters had long-term persistence with evidence of at least almost two decades of local persistence between the estimated time of introduction and the most recent sampling. In OR, most clusters were estimated to have occurred more recently (within 3 years), but two clusters showed at least a decade of persistence (Fig. 6). Complex clusters, although exhibiting a higher median persistence time of 6.2 years compared to dyads (4.2), did not show a statistically significant difference ($P = 0.1359$, Mann–Whitney test).

### Factors associated with transmission clusters

In unadjusted logistic regression analyses, membership in a cluster was associated with younger age (21% vs. 30%; OR = 2.54 [95% CI: 1.41–4.60], $P = 0.002$; Table 3), being recruited by a partner, spouse, boyfriend or girlfriend (10% vs. 19%; OR = 2.07 [95% CI: 1.17–3.67], $P = 0.013$; Table 3), and having an illegal source of income (e.g., selling drugs, selling sex and theft) (27% vs. 39%; OR = 1.75 [95% CI: 1.14–2.67], $P = 0.010$; Table 3). A borderline significant association was found with those who have shorter incarcerated period and clustering (21 vs. 14 days; OR = 1.01 [95% CI: 1–1.01], $P = 0.05$; Table 3) while receiving income assistance (e.g., disability check, military, TANF, AFDC) (29% vs. 16%; OR = 0.477 [95% CI: 0.29–0.80], $P = 0.005$; Table 3) decreased the odds of being in a cluster. No significant differences were found between HCV subtype, race and ethnicity, education, drug choice or frequency of injection drug use. The multivariable model included factors that were associated with clustering ($P < 0.10$) in the

univariable analysis, including age, source of recruitment, incarceration time, income source and syringe source. The factors that remained significantly associated with membership in a cluster was being aged 18–29 years (AOR = 2.009 [95% CI: 1.17–3.46], $P = 0.012$; Table 3). In contrast, receiving a form of public assistance as a primary income source remained negatively associated with clustering (AOR = 0.544 [95% CI: 0.30–0.97], $P = 0.040$; Table 3). However, we recognize that conventional outcome-based variable selection may overfit the data. To address this, we conducted a penalized logistic regression with cross-validation and bootstrap stability assessment (Supplementary Table 2). The penalized logistic regression achieved a mean cross-validated ROC-AUC of 0.60 (SD 0.08), indicating modest discrimination for cluster membership. After shrinkage, most candidate predictors were reduced to zero. A small number of variables were consistently retained, including younger age (18–29 years vs 44–65), methamphetamine (vs opioids) as drug of choice, recruitment through partner/spouse or peers, illegal income source and syringe acquisition from drug dealers. Effect sizes were modest, with odds ratios close to 1, and bootstrap selection frequencies ranging from ~55% to 86%.

## Discussion

Despite the high incidence of HCV among PWUD in recent years and the ensuing opioid overdose crisis, very little has been understood about the emergence and spread of the virus across the United States. Rural communities have been disproportionately affected by HCV outbreaks, fueled by overlapping epidemics or syndemics of injection drug use, widespread nonmedical use of opioids and stimulants, and compounded by social inequities and social barriers[19–21]. While HCV screening rates among PWUD range from 8 to 32%[22,23] these rates are markedly lower in rural areas with some estimates as a lows as 6%[23]. In this multi-site cohort study comprising 692 PWUD in the rural United States, we found that nearly one-third of all HCV infections were genetically linked, with genotype 1a predominating across all study sites. Subtle differences in genotype distribution were observed between sites, with Kentucky and New England showing the greatest heterogeneity, including the presence of less common subtypes such as 4a and 2a which may lead to less than optimum treatment outcomes[24]. The prevalence of mixed HCV genotype infections, as determined by sequencing, was 4.8% - this rate is consistent with other studies reporting low frequencies[25–29]. Although a higher proportion (18%) of mixed-strain HCV infections was reported in an outbreak in rural Indiana using the GHOST platform[11], such findings have not been

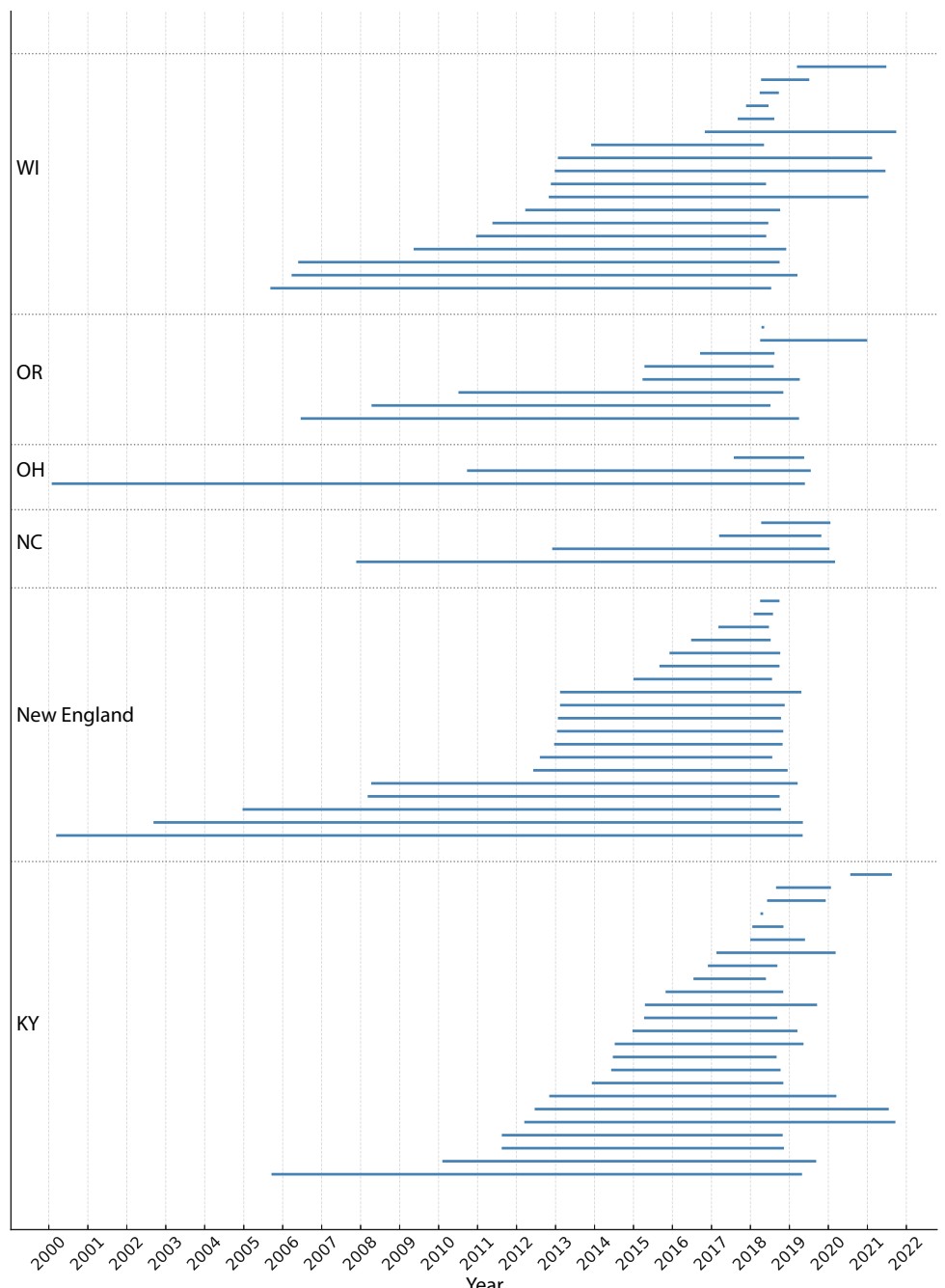

**Fig. 6 | Timescale of persistence of each detected transmission cluster by study site.** Each horizontal line represents a genomic cluster of HCV sequences showing the timespan from the inferred introduction time of the cluster to the latest sampled sequence. Clusters are grouped by study sites (Wisconsin, Oregon, Ohio, North Carolina, New England, and Kentucky) and separated by dotted horizontal lines and labeled along the vertical axis. The *x*-axis indicated calendar year, allowing for a comparison of persistence times across states and over time. Note that some clusters may not be present as this only represents those clusters found using a timescaled phylogenetic tree of the consensus sequence data. Other clusters, such as minor variant clusters detected from the analyses of deep sequencing data from Global Hepatitis Outbreak Surveillance Technology, are not depicted here. The New England study site is composed of Vermont, New Hampshir,e and Massachusetts.

replicated in other rural settings and are likely due to the rapid transmission dynamics and unique social structure of that specific community. While the GHOST pipeline can identify mixed infections it should be noted that the amplicon size and the high rate of evolution may limit resolution in distinguishing true mixed infections within the same genotype without complementary deep sequencing of another genomic region. Further, comparisons across studies are challenging due to differences in cohort selection, injection behaviors and the genomic region analyzed[30].

The degree of clustering observed within this study at 29.5% is similar to that observed from other injecting drug cohorts. For example, a study examining HCV transmission across four Indian cities revealed that 28.8% of HCV sequences clustered[31], while studies conducted in North America have revealed variable levels of clustering from 46% in Baltimore[15], 33% in Wisconsin[18], 25% in New York[32], 31% in Vancouver[33] and 36% in Ottawa[34]. Higher rates of clustering were observed in those studies that enrolled injecting partnerships, with 54% of Australian participants genetically related[35] compared with 52%

**Table 3 | Crude and multivariable logistic regression analysis of factors associated with being in a cluster for participants in the Rural Opioid Initiative**

| Characteristic | No cluster (n = 286) | Cluster (n = 143) | Unadjusted OR[a] (95% CI[b]) | P value | Adjusted OR[a] (95% CI[b]) | P value |
|---|---|---|---|---|---|---|
| Age groups | | | | | | |
| 18–29 | 69 (20.7%) | 43 (30.1%) | 2.544 (1.407, 4.599) | 0.002 | 2.009 (1.166, 3.462) | 0.012 |
| 30–35 | 67 (23.5%) | 33 (23.1%) | 1.590 (0.852, 2.969) | 0.146 | | |
| 36–43 | 71 (24.9%) | 39 (27.3%) | 1.966 (1.071, 3.610) | 0.029 | 1.630 (0.938, 2.832) | 0.083 |
| 44–65 | 88 (30.9%) | 28 (19.6%) | Ref | | | |
| Gender | | | | | | |
| Female (vs. male) | 100 (35.0%) | 55 (38.5%) | 1.162 (0.767, 1.761) | 0.477 | | |
| Recruitment source | | | | | | |
| Friend, associate, acquaintance | 163 (57.0%) | 79 (55.2%) | Ref | | | |
| Partner, spouse, boyfriend, girlfriend | 29 (10.1%) | 27 (18.9%) | 2.069 (1.169, 3.663) | 0.013 | 1.736 (0.926, 3.254) | 0.085 |
| Casual sex partner | 5 (1.8% | 3 (2.1%) | 1.108 (0.259, 4.745) | 0.890 | | |
| Family member | 23 (8.0%) | 7 (4.9%) | 0.585 (0.244, 1.400) | 0.228 | | |
| Neighbor | 4 (1.4%) | 2 (1.4%) | 1.195 (0.281, 5.078) | 0.809 | | |
| Person I use drugs with | 34 (11.9%) | 25 (17.5%) | 1.014 (0.233, 4.411) | 0.125 | | |
| Service or program staff | 5 (1.8%) | 4 (2.8%) | 1.612 (0.426, 6.104) | 0.482 | | |
| Stranger | 3 (1.1%) | 1 (0.7%) | 0.662 (0.068, 6.422) | 0.722 | | |
| Other | 3 (1.1%) | 2 (1.4%) | 1.333 (0.220, 8.076) | 0.754 | | |
| Race and ethnicity | | | | | | |
| Non-Hispanic White | 240 (83.9%) | 124 (86.7%) | Ref | | | |
| Non-Hispanic Black | 7 (2.4%) | 1 (0.7%) | 0.310 (0.035, 2.707) | 0.289 | | |
| Non-Hispanic American Indian | 14 (4.9%) | 6 (4.2%) | 1.055 (0.333, 3.343) | 0.928 | | |
| Non-Hispanic Other | 10 (3.5%) | 4 (2.8%) | 0.951 (0.260, 3.546) | 0.951 | | |
| Hispanic | 15 (5.2%) | 8 (5.6%) | 1.503 (0.500, 4.514) | 0.468 | | |
| Education | | | | | | |
| High school diploma or GED[c] | 143 (50.2%) | 69 (48.3%) | Ref | | | |
| Some college | 73 (25.6%) | 41 (28.7%) | 1.91 (0.678, 2.093) | 0.543 | | |
| Less than high school | 63 (22.1%) | 31 (21.7%) | 0.915 (0.519, 1.615) | 0.760 | | |
| College graduate or above | 6 (2.1%) | 2 (1.4%) | 0.630 (0.124, 3.200) | 0.577 | | |
| Median duration of recent jail or prison time (days) | 20.5 | 14 | 1.005 (1.00, 1.011) | 0.050 | 1.002 (0.997, 1.008) | 0.437 |
| Homeless, past 6 months | 175 (61.2%) | 85 (59.4%) | 0.928 (0.613, 1.404) | 0.724 | | |
| Drug class of choice | | | | | | |
| Opioids | 150 (52.5%) | 96 (67.1%) | Ref | | | |
| Fentanyl | 11 (3.9%) | 4 (2.8%) | 1.057 (0.320, 3.49) | 0.927 | | |
| Methamphetamine | 89 (31.1%) | 29 (20.3%) | 0.851 (0.428, 1.690) | 0.644 | | |
| Cocaine | 15 (5.2%) | 8 (5.6%) | 1.655 (0.652, 4.198) | 0.289 | | |
| Benzodiazepines | 1 (0.4%) | 0 | — | — | | |
| Other | 10 (3.5%) | 3 (2.1%) | 0.859 (0.226, 3.26) | 0.824 | | |
| Medications for opioid use disorder | 10 (3.5%) | 3 (2.1%) | 0.859 (0.226, 3.26) | 0.824 | | |
| HIV co-infection | 2 (0.7%) | 3 (2.1%) | 6.110 (0.630, 59.289) | 0.119 | | |
| Health Insurance | 219 (76.7%) | 111 (77.6%) | 1.068 (0.645, 1.769) | 0.798 | | |
| Injecting drug use frequency | | | | | | |
| Daily or more | 200 (74.6%) | 97 (72.4%) | Ref | | | |
| More than weekly but no daily | 26 (9.7%) | 14 (10.5%) | 0.983 (0.431, 2.244) | 0.968 | | |
| Weekly | 17 (6.3%) | 9 (6.7%) | 0.964 (0.380, 2.444) | 0.939 | | |
| Monthly | 25 (9.3%) | 14 (10.5%) | 1.047 (0.458, 2.395) | 0.913 | | |
| Income source | | | | | | |
| Illegal | 77 (26.9%) | 56 (39.2%) | 1.747 (1.142, 2.674) | 0.010 | 1.444 (0.887, 2.351) | 0.140 |
| Legal | 169 (59.1%) | 96 (67.1%) | 1.414 (0.928, 2.155) | 0.107 | | |
| Income assistance | 82 (28.7%) | 23 (16.1%) | 0.477 (0.285, 0.798) | 0.005 | 0.544 (0.304, 0.972) | 0.040 |
| Source of needles | | | | | | |
| SSP/NEP[d] | 103 (38.0%) | 54 (40.6%) | Ref | | | |
| Pharmacy | 60 (22.1%) | 34 (25.6%) | 1.36 (0.789, 2.348) | 0.270 | | |

**Table 3 (continued) | Crude and multivariable logistic regression analysis of factors associated with being in a cluster for participants in the Rural Opioid Initiative**

| Characteristic | No cluster (*n* = 286) | Cluster (*n* = 143) | Unadjusted OR[a] (95% CI[b]) | P value | Adjusted OR[a] (95% CI[b]) | P value |
|---|---|---|---|---|---|---|
| Someone else who got them from an SSP/NEP[d] | 33 (12.2%) | 22 (16.5%) | 1.579 (0.848, 2.940) | 0.150 | | |
| Drug dealer | 23 (8.5%) | 4 (3.0%) | 0.337 (0.112, 1.007) | 0.052 | 0.341 (0.112, 1.038) | 0.058 |
| Spouse, partner or relative, friend | 49 (18.1%) | 18 (13.5%) | 0.716 (0.385, 1.334) | 0.294 | | |
| Other | 3 (1.1%) | 1 (0.8%) | 0.705 (0.722, 6.888) | 0.764 | | |
| HCV subtype | | | | | | |
| 1a | 84 (58.7%) | 139 (62.6%) | Ref | | | |
| 1b | 4 (2.8%) | 7 (2.5%) | 1.143 (0.77, 16.947) | 0.923 | | |
| 2a | 2 (1.4%) | 1 (0.4%) | 4.000 (0.134, 119.230) | 0.423 | | |
| 2b | 7 (4.9%) | 23 (8.0%) | 0.609 (0.048, 7.758) | 0.702 | | |
| 3a | 45 (31.5%) | 74 (25.9%) | 1.216 (0.107, 13.799) | 0.874 | | |

Univariable and multivariable logistic regression (two-sided) were performed to assess associations with transmission cluster membership. Exact *P* values are shown. No formal correction for multiple testing was applied. Bold values indicate statistically significant values. ($P \leq 0.05$).
[a]*OR* odds ratio.
[b]*CI* confidence intervals.
[c]*GED* general education development.
[d]*SSP* Syringe Services Programs, *NEP* Needle Exchange Programmes.

of injecting partnerships from San Francisco[17]. The elevated clustering rate observed in these known partnerships may be attributed to the study designs, which involves more frequent assessments, thereby increasing the likelihood of capturing transmission events early. Differences in the rate of clustering among participants could reflect regional differences in drug use networks, recruitment approaches, sampling density, clustering analyses methods or behavioral differences. It may also be attributed to the use of different amplification strategies involving different primer sets and target regions. Age has been previously demonstrated to be an important factor in HCV transmission, with higher rates of clustering found in participants of younger age[36]. It should also be noted that the transmission clusters identified in this study are most likely to represent recent transmission events. HVR1 provides strong discriminatory power to distinguish closely related infections, but its high mutation rate means that infections linked by older or more distant transmission chains may have accumulated substantial divergence and may not be genetically related. Thus, the clustering we observe should be interpreted primarily as evidence of ongoing or recent transmission, rather than reflecting longer-term or historical transmission networks.

During the acute and early stages, the viral population tends to be relatively homogeneous, primarily due to serial bottlenecks and the presence of a single founder virus. As the infection progresses, HCV undergoes increased genomic diversification as it adapts to the host's immune response, leading to a positive correlation between the stage of infection and intra-host viral diversity. If a correlation does exist then genetic diversity may be used as a proxy for infection recency as previously suggested[37–40], then this may well suggest that clusters detected in this study are more driven by individuals who are in the earlier stages of infection, when genomic diversity is relatively homogeneous owing to a genetic bottleneck effect. We found that 49% of transmission clusters stemmed from the same social recruitment chains, suggesting that this type of recruitment approach is successful in uncovering transmission networks among PWUD. The time of persistence of these local transmission clusters was estimated from dated phylogenies and revealed prolonged period of persistence over 10 years in some geographical locations, with more complex transmission networks cryptically spreading for longer periods compared to those in a dyadic cluster.

Younger age (particularly those aged 18–29) emerged as the strongest independent predictor of transmission cluster membership, consistent with previous findings that highlight elevated transmission

risk among younger PWUD[36]. Recruitment by a sexual or intimate partner and having an illegal source of income were also associated with increased odds of clustering in unadjusted analyses, suggesting that close personal networks and high-risk socioeconomic behaviors may be involved in viral spread. Sexual relationships have also been associated with increased sharing of syringes and injecting equipment and having a genetically related infection in young PWUD from San Francisco[17,41,42]. Conversely, receiving public assistance as a primary income source was negatively associated with clustering, potentially reflecting reduced engagement in higher-risk networks. No associations were found between clustering and HCV subtype, race and ethnicity, education level, drug type, or frequency of injection, underscoring the importance of social and structural factors rather than strictly behavioral factors in shaping transmission dynamics.

From this study, we have shown how genomic sequencing has the potential to provide a high-resolution picture of HCV evolution and transmission, providing public health authorities with actionable information. Molecular epidemiology methods have been used in related fields (e.g., HIV prevention) for many years and have been critical to informing public health interventions. A striking example of this approach was from an implementation case study in Canada that used phylogenetic analysis of routine clinical data and demonstrated that "near-real-time" analysis directly impacted ongoing viral transmission and led to enhanced public health follow-up with linkage to care and treatment initiation[43]. As part of the United States Federal Ending the HIV Epidemic strategic initiative, rapidly detecting and responding to emergent clusters of HIV infection is one of the key pillars that will be used to further reduce new transmissions[44]. Yet, its implementation for HCV remains scarce but there is enormous potential for genomic surveillance to augment traditional surveillance approaches. It is especially valuable for marginalized population groups (i.e., people who use injection drugs), where it could be used for targeted network-based strategies to interrupt transmission[45], which could aid in microelimination strategies[46]. Moreover, dried blood spots have also been validated for genomic surveillance as a less invasive alternative to venous blood draws[32,47] and could serve as a useful tool for future surveillance efforts.

This study has several limitations. Firstly, recruiting marginalized and highly stigmatized populations, such as people who inject drugs, poses significant challenges, especially in rural settings where no universal recruitment tool exists. In this study, a modified form of chain referral sampling, RDS, was utilized to recruit PWUDs. Although RDS

has not been widely adopted in rural areas, it has been successfully implemented in multiple rural U.S. regions[48,49]. While there were some differences in recruitment across study sites, little variation was observed when multiple variables were assessed. Although each site was instructed to collect and process specimens in the same manner, there may have been some time delays to centrifugation and freezing of samples, which may have compromised the integrity of the samples leading to different PCR failure rates across sites. Secondly, the study focused on a fragment within the E1/2 region of the HCV genome. While this captures only a small portion of the HCV genome, it was selected for its high variability, making it effective for detecting recent transmission events in outbreak settings[50]. Older or more distant links may have been missed due to sequence divergence in the region. We acknowledge that amplicon-based sequencing can introduce biases that may confound viral diversity estimates; however, these were mitigated using well-validated primers, high-fidelity polymerases, and the GHOST pipeline's quasispecies-based error correction. Nonetheless, sequencing of additional genomic regions, such as NS3 and NS5A/B, would provide complimentary insights particularly for monitoring antiviral drug resistance patterns. Thirdly, genetic clustering by similarity does not confirm a transmission event, as there may be unsampled missing links within transmission chains, leading to an incomplete understanding of the HCV transmission network. Additionally, the direction of HCV transmission cannot be readily inferred from genetic data alone[17], though phylogenetic analysis can provide insights into transmission direction with varying degrees of reliability[51–53]. Finally, behavioral data were self-reported and may be subject to recall and response biases.

In conclusion, this study of PWUD across multiple U.S. rural sites reveals that the HCV epidemic is not uniform across regions, with notable differences in genotype diversity, clustering rates, intra-host viral diversity, and the persistence of transmission clusters. These findings suggest that local HCV epidemics are evolving at different stages and highlight the importance of robust genomic surveillance to guide the development of targeted social and structural public health interventions for high-risk, marginalized rural communities.

## Methods

### Study participants

A cross-sectional survey of people who used drugs in rural counties with high overdose rates from ten U.S. states and 66 U.S. counties (Illinois, Kentucky, North Carolina, New England [Massachusetts, New Hampshire, and Vermont], Ohio, Oregon, West Virginia, and Wisconsin) was conducted and herein referred to as the Rural Opioid Initiative (ROI). Additional details on the ROI consortium have been previously published[14]. Kentucky samples were largely received from another study, termed Social Networks Among Appalachian People (SNAP), but participants were recruited in the same manner and timeframe. Study participants were recruited between January 2018 and December 2021. Individuals were eligible for inclusion if they lived in the study area, reported any past 30-day injection drug use and/or noninjecting opioid use "to get high" (heroin, prescription pain medication). Inclusion criterion for all sites was age ≥18 years except two states (Illinois, Wisconsin), where the age criterion was ≥15 years. All sites conducted recruitment using respondent-driven sampling to facilitate sampling of hard-to-reach populations. Each study site identified "seed" participants to initiate recruitment chains. Seeds were recruited from syringe service programs, local health departments, and community outreach to represent the general demographic characteristics of the local eligible population. Seeds recruited up to six members of their drug use network. Each referred participant recruited their network peers similarly with the goal of maximizing recruitment chains. Participants received $10 to $20 per successfully enrolled peer and $40–$60

for completion of study procedures. A complete evaluation of respondent-driven sampling in this setting has been published elsewhere[54]. The gender of participants was determined based on self-reporting. Blood specimens were collected for rapid HIV, HCV, and syphilis testing at all sites. Specimens that were positive for HCV antibodies were shipped to the GHOST (Global Hepatitis Outbreak Surveillance Technology) Sequencing Center at the Ragon Institute of Massachusetts General Hospital, Massachusetts Institute of Technology, and Harvard for RNA testing and viral genomic analysis.

### Nucleic acid extraction and PCR amplification

RNA was isolated from 140 µl of plasma using the QIAamp Viral RNA Mini Kit (Qiagen, Hilden, Germany). A one-step RT-PCR reaction was performed on all samples to amplify a segment at the E1/E2 junction of the HCV genome, which contains the hypervariable region 1 (HVR1) due to its high variability and its ability to reliably detect transmission events in outbreak settings[50]. The first round RT-PCR consisted of an Illumina adapter specific portion, a sample specific barcode segment, and an HCV HVR specific primer segment, F1- GTGACTGGAGTTCA-GACGTGTGCTCTTCCGATCT-NNNNNNNNNN-GGA-TAT-GAT-GAT-GAA-CTG-GT and R1-ACA-CTC-TTT-CCC-TAC-ACG-ACG-CTC-TTC-CGA-TCT-NNNNNNNNNN-ATG-TGC-CAG-CTG-CCG-TTG-GTG-T at a final concentration of 4 pM amplified using Superscript III RT/Platinum Taq DNA Polymerase High Fidelity with the following conditions: cDNA synthesis for 30 min at 55 °C, followed by heat denaturation at 95 °C for 2 min, the PCR amplification conditions were 40 cycles of denaturation (94 °C for 10 s), annealing (55 °C for 10 s) and extension (68 °C for 10 s) with a final extension at 68 °C for 5 min. Amplified products were run on a 1% agarose gel and either PCR purified with the QIAquick PCR purification kit (Qiagen) or gel extracted and purified using the PureLink quick gel extraction kit (Invitrogen). A second round limited cycle PCR (94 °C for 2 min, (94 °C for 15 s; 55 °C for 30 s; 68 °C for 30 s) × 8 cycles, 68 °C for 5 min) is performed to add barcode specific indexes and sequencing specific adapters and primers to each sample to allow for multiplexing as well as internal controls for cross-contamination. Negative controls were introduced at each stage of the procedure and all PCR procedures were performed under PCR clean room conditions using established protocols. Indexed samples are 0.7× SPRI purified two times to remove excess primer dimer and short fragments that can interfere with the sequencing process. To avoid contamination, all reagents were pre-aliquoted with dedicated equipment ensuring physical separation of sample processing from pre- and post-PCR amplification steps including deep sequencing.

### Illumina deep sequencing

PCR amplicons of size 306 bp were quantified using the Picogreen kit (Invitrogen, Carlsbad, CA) on a Fluorometer ST (Promega, Madison, WI) with the integrity of the fragment evaluated using a Bioanalyzer 2100 (Agilent, Santa Clara, CA). Samples were pooled and sequenced on an Illumina MiSeq platform using a 2 × 250 bp V2 Nano reagent kit. In general, a sequence library consisted of between 8 and 16 specimens including one negative control for every 7 serum specimens.

### Deep sequencing data analysis

Sequencing reads were automatically demultiplexed and duplicate reads were removed using fastuniq v1.1[55] to limit the influence of PCR artifacts and subsequently quality trimmed using trimmomatic v0.36[56] if sequencing adapters or low-quality bases (Phred scores <20) were detected. Vicuna v1.1, a de novo consensus assembly algorithm[57], was used to generate consensus assemblies from genetically heterogeneous populations with automated computational finishing and annotation of de novo viral assemblies performed using V-FAT v1.1 as previously performed[58]. All consensus assemblies were cross checked with the iVar pipeline[59] and any differences were further inspected.

Lastly, intra-host single nucleotide variants were identified with V-Phaser 2[60]. Instrain was used to assess the genomic nucleotide diversity ($\pi$) based on all reads and calculated as the average number of nucleotide differences per base pair[61].

## HCV genotyping

All de novo consensus sequences were classified using the method implemented in the Genome Detective virus tool for phylogenetic genotyping[62]. To detect and quantify the presence of mixed HCV genotypes in Illumina MiSeq sequencing reads paired end reads were mapped using bowtie2 to a set of HCV reference genome sequences ($n = 571$). The absolute number of reads that mapped to a reference genome of the given HCV genotype was then counted and quantified. The criteria for identifying mixed genotype infections were that it had to be at a frequency of 1% or greater and have at least 200 reads mapped to it. This cutoff was seen as a conservative threshold to exclude cross-contamination between samples.

## Phylogenetic reconstruction

All de novo consensus sequences were aligned using MAFFT v7.470[63] and IQ-TREE v 2.1[64] was used to construct a maximum likelihood phylogenetic tree employing the best-fit model of nucleotide substitution according to the Bayesian Information Criterion as indicated by the Model Finder application implemented in IQ-TREE[65]. Statistical robustness of individual nodes was determined using 1000 ultrafast bootstrap replicates[66].

## GHOST analysis

Global Hepatitis Outbreak and Surveillance Technology (GHOST) detects and visualizes transmission clusters using deep sequencing data from the hypervariable region of the HCV genome. Paired-end reads for each successful sequenced sample were uploaded to the GHOST server where the data were subjected to quality control before being analyzed for the presence of transmission links using hamming distance. Two cases were considered linked by transmission if the distance between them was less than the empirically defined threshold value of 0.037. Further details on GHOST have been published by Longmire et al.[67].

## Estimation of time of the most recent ancestor

To characterize the time of the most recent ancestor (tMRCA) of transmission clusters we used the phylogenetic framework as implemented in the Nextstrain pipeline[68]. As our range of sample dates were not large or wide enough, we did not have sufficient temporal signal as measured by the coefficient of the root-to-tip regression method. In line with prior HCV studies, we employed an independent dataset with significant temporal information to provide the substitution rates of the genomic region of interest[69]. The SRD06 nucleotide partitioned-substitution model, an uncorrelated relaxed lognormal molecular clock and a Bayesian skyline coalescent model with 10 groups was employed using BEAST version 1.10.4[70] from which we obtained rate estimates for the precise subgenomic region sequenced in this analysis. The Markov chain Monte Carlo chains (MCMCs) were run for 500 million generations and sampled regularly to yield a posterior tree distribution based upon 10,000 estimates. The mean rate of evolution and standard deviation of the estimated rate of evolution were then used in the Nextstrain workflow to infer the time scale of HCV clusters.

## Statistical analyses

Descriptive statistics to summarize demographic characteristics by sample status and sequenced sample status were calculated. Differences in categorical variables were assessed using chi-square tests, while continuous variables were compared using t-tests for sample status and one-way analysis of variance (ANOVA) for overall comparisons. Logistic regression analyses were used to identify factors associated with being in a dyad or cluster (yes/no). Univariate logistic analysis was first performed, and variables with $p < 0.10$ were selected for multivariable logistic regression analysis. All tests were two-tailed and a $p$-value $< 0.05$ was considered statistically significant. To limit overfitting and avoid outcome-driven variable selection, we fitted an L1-penalized (LASSO) logistic regression including all prespecified covariates from Table 3: age group, sex, recruitment source, race/ethnicity, education, homelessness, drug class of choice, HIV status, health insurance, injection frequency, income source, and syringe source. Age was categorized into four groups (18–29, 30–35, 36–43, and 44–65 [reference]); categorical variables were one-hot encoded with appropriate reference categories; binary indicators were coded as 0/1. Predictors were standardized before modeling. The tuning parameter ($\lambda$) was selected by 10-fold cross-validation maximizing the area under the ROC curve (AUC). To assess the robustness of selected predictors, we performed nonparametric bootstrap resampling ($B = 200$), refitting the penalized model in each resample and calculating the frequency with which each predictor was retained (bootstrap selection frequency). For interpretability, we subsequently refitted an unpenalized logistic regression including only the LASSO-selected predictors to estimate odds ratios (ORs) and 95% confidence intervals (CIs). All analyses were performed using Python and Stata software (version 18.0; StataCorp, College Station, Texas, USA).

## Ethics statement

All participants provided written informed consent prior to sample or data collection at all participating institutions. All study procedures and protocols were approved by the Institutional Review Board at each participating site, which were also reviewed and approved by the Institutional Review Board of Massachusetts General Hospital.

## Reporting summary

Further information on research design is available in the Nature Portfolio Reporting Summary linked to this article.

## Data availability

Due to the sensitive nature of the topic area the current analyzed dataset is not publicly available. However, Rural Opioid Initiative (ROI) data are available on request and may be subject to a data use agreement. Please see the ROI website (https://ruralopioidinitiative.org) for more information on how to complete a concept proposal and data request. Please contact the ROI consortium (ruralopioids@uw.edu) for more information or for data requests. The consensus sequence generated in this study have been deposited to GenBank under accession numbers PX261275 - PX261966.

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

## Acknowledgements

In memoriam, we acknowledge the contributions of T.M.A. who passed away unexpectedly before publication. T.M.A. contributed to the conceptualization, funding, and design of this study and provided critical input on data interpretation. His insights and expertise was instrumental in shaping this research. This work was supported by the National Institute of Drug Abuse (NIDA) grant U24DA044801 (T.M.A.). Data is based upon data collected and/or methods developed as part of the Rural Opioid Initiative (ROI), a multi-site study with a common protocol which was developed collaboratively by investigators at eight research institutions and at the National Institute of Drug Abuse (NIDA), the Appalachian Regional Commission (ARC), the Centers for Disease Control and Prevention (CDC), and the Substance Abuse and Mental Health Services Administration (SAMHSA). Primary data collection was supported by grants UG3DA044798 (A.M.Y., H.C.), UG3DA044830 (P.D.F.), UG3DA044831 (P.T.K.), UG3DA044826 (R.P.W.) co-funded by NIDA, ARC, CDC, and SAMHSA. Specimen collection in Kentucky was also supported by R01DA033862 (J.R.H.) and R01DA047952 (J.R.H.). The authors thank the other ROI investigators and their teams, the ROI Executive Steering Committee chair, Dr. Holly Hagan, the NIDA Science Officer, Dr. Richard Jenkins, and, particularly, the participants of the individual ROI studies for their valuable contributions. We also thank Yury Khudyakov and the team at the CDC Division of Viral Hepatitis for supporting the GHOST work.

## Author contributions

D.C.T. and T.M.A. acquired funding, conceived, designed, and supervised the study. D.J.B., J.S., T.L.N., and K.A.P. performed all experiments. D.C.T. analyzed all data and prepared the figures. D.B., H.C., J.F., P.D.F., K.R.H., J.R.H., S.B., C.H., W.J., P.T.K., W.M., M.T.P., G.S., T.J.S., R.P.W., and A.M.Y. contributed to the review and editing process and provided funding acquisition. J.I.T. and L.S.M. were involved in data curation and contributed to the review and editing process. D.C.T. and T.M.A. wrote the manuscript. All authors read and approved the final manuscript.

## Competing interests

The authors declare no competing interests.

## Additional information

[1]Medical Research Council/Uganda Virus Research Institute and LSHTM Uganda Research Unit, Entebbe, Uganda. [2]Department of Infectious Disease Epidemiology, London School of Hygiene and Tropical Medicine, London, UK. [3]Ragon Institute of MGH, MIT and Harvard, Cambridge, MA, USA. [4]Division of Epidemiology, College of Public Health, Ohio State University, Columbus, OH, USA. [5]Department of Behavioral, Social, and Health Education Sciences, Rollins School of Public Health, Emory University, Atlanta, GA, USA. [6]Departments of Behavioral Medicine and Psychiatry and Medicine/Infectious Diseases, West Virginia University School of Medicine, Morgantown, WV, USA. [7]University of Massachusetts Chan Medical School - Baystate and Baystate Health, Springfield, MA, USA. [8] Friends Research Institute, Baltimore, ML, USA. [9]Department of Behavioral Science, Center on Drug and Alcohol Research, University of Kentucky College of Medicine, Lexington, KY, USA. [10]Institute for Global Health and Infectious Diseases, University of North Carolina at Chapel Hill, Chapel Hill, NC, USA. [11]Department of Public Health Sciences, Clemson University, Clemson, SC, USA. [12]Oregon Health & Science University, Portland, OR, USA. [13]The Ohio State University, Columbus, OH, USA. [14]Section of Infectious Diseases and Global Health, Department of Medicine, University of Chicago, Chicago, IL, USA. [15]School of Public Health, West Virginia University, Morgantown, WV, USA. [16]Department of Public Health and Community Medicine, Tufts University School of Medicine, Boston, MA, USA. [17]Department of Medicine, School of Medicine, University of Washington, Seattle, WA, USA. [18]Division of Infectious Diseases, Department of Medicine, University of Wisconsin School of Medicine and Public Health, Madison, WI, USA. [19]Department of Epidemiology and Environmental Health, University of Kentucky, Lexington, KY, USA. ✉e-mail: damien.tully@lshtm.ac.uk

