## [Peer Review file · Nature Communications]

Genomic surveillance uncovers regional variation in HCV transmission networks in rural United States

Corresponding Author: Dr Damien Tully

Version 0:

Reviewer comments:

Reviewer #1

(Remarks to the Author)

Whilst I appreciate that the authors used a high-fidelity polymerase for their PCR reactions, I would be slightly concerned about the error rate in the sequencing after putting the samples through 48 cycles of PCR plus Illumina sequencing which also requires additional amplification. Was there any data to indicate this was not an issue?

It would be useful to provide the size of the HVR1 amplicon generated with the PCR as it is helpful to understand for several of the analytical steps.

An 18% failure rate for a PCR is very high if samples are known to be RNA positive – was the PCR used in this study well validated in previous work? Highly sensitive PCRs can be developed for the E1/E2 region, so this failure rate is concerning. Data availability – are the authors planning to submit consensus sequences to Genbank or another appropriate data repository? The reasons for not doing this are unclear to me, and sequences can be held under embargo prior to publication of the manuscript if required.

Figure 1 – unclear where the genotype distribution data is in the figure?

The authors state that – “Almost one-fifth (19%) cleared HCV infection with treatment.”. Where were they collecting this data from, were participants followed up at a later date? Does this mean four-fifths did not receive treatment or failed treatment?

Figure 2 – did the authors find any evidence of mixed infections of the same genotype or was the approach used only able to identify mixed genotype/subtype infections? I would expect these to also be present among active PWUD?

Figure 3 – Presentation of this tree could be improved, as the gt1b clade appears to be in the middle of the gt1a sequences? Based on the number of sequences, I assume the sequences below the gt1b cluster are gt1a but this is not clear and should be more clearly indicated. Again amplicon length would be helpful here and some evidence of bootstrapping. How were consensus sequences derived for mixed gt infections?

GHOST analysis – the authors should provide some more details here. Were these samples all one particular genotype or are multiple genotypes shown in the figure? Presumably linkage between different genotypes is not possible, so this is helpful to understand. More information on what the single dots would be helpful – are these unlinked samples? Also the authors used an empirically defined threshold value of 0.037 to define linkage – did the authors consider doing any sensitivity analysis on this (ie if this is adjusted slightly do you see more linkage?)

I think the analysis on viral diversity is confounded by the use of amplicons for deep sequencing – as the method to obtain the sequences can introduce errors into the sequences, so the authors need to be more nuanced in their conclusions in this section and discuss this as a limitation.

Table 1 – I would encourage the use of some statistics here to demonstrate the similarity between the subsets of the cohorts. The raw data is insufficient to illustrate this?

Reviewer #2

(Remarks to the Author)

This is a cross-sectional study conducted among people who used drugs (PWUD) in rural counties, recruited between 2018 and 2021. The study includes individuals residing in the study area who reported any past 30-day injection drug use and/or non-injecting opioid use “to get high” (including heroin and prescription pain medications). Participants were recruited using respondent-driven sampling to characterize the circulating HCV strains circulating among PWUD in rural US areas and to identify factors associated with HCV transmission clusters. The results include a description of the prevalence of HCV subtypes in the rural PWUD population and their phylogenetic clustering, having a very interesting discussion.

Comments:

While the study addresses an important public health issue, it lacks elements of novelty that would significantly enhance its contribution to the field. To bring added value to this type of molecular epidemiology work, it would have been highly informative to include detailed epidemiological data linking participants, such as information on shared injection practices, paraphernalia, or other risk behaviors. Establishing clearer epidemiological relationships between participants and the identified transmission clusters is crucial to strengthen the interpretation of the phylogenetic findings.

In addition, given the inherent intra-host variability of HCV, phylogenetic analysis using consensus sequences may not adequately capture all transmission relationships particularly when attempting to define clusters among PWUD-related HCV transmissions. This limitation is especially relevant due to the potential bottlenecks in transmissions, where a subpopulation of the viral quasispecies is passed from one individual to another.

Furthermore, analyzing multiple genomic regions with differing levels of variability (e.g., a highly variable region such as E1/E2 or HVR1, an intermediate like NS5B, and a more conserved region like core) would have improved the resolution and robustness of the phylogenetic inferences.

If the transmission event occurred recently, phylogenetic analysis using the highest variable region has strong discriminatory power to infer relatedness between samples, generating a clear transmission cluster. However, if the transmission occurred several months or years ago, the quasispecies in each host may have diverged significantly, requiring the study of a viral genomic region with median or lower variability to detect phylogenetic links.

While the authors correctly target a relatively conserved region within the E1/E2 region of the HCV genome, their use of a single primer pair, without degenerate positions, raises concerns about potential amplification bias across subtypes.

Although this region is overall conserved, several subtype-specific nucleotide polymorphisms are well documented and can significantly impact primer annealing efficiency.

In the forward primer (F1: 5'-GGATATGATGATGAACTGGT-3'; positions 1356–1375), position 1359 is often a C in most genotype 1 (G1) sequences, whereas the primer sequence includes a T, potentially reducing amplification efficiency in G1 viruses. Additionally, genotypes 2b and 3 commonly present two or more mismatches within this region, and genotype 4 strains often carry a mismatch at the 3'-end (position 1375), where they consistently have an A, further impairing amplification. Notably, the primer may favor amplification of G3 viruses, which typically retain a T at position 1359, resulting in overrepresentation of this subtype in downstream analyses.

The reverse primer (R1: 5'-ATGTGCCAGCTGCCGTTGGTGT-3'; positions 1652–1673), although designed in a relatively conserved region as well, also contains subtype-specific mismatches. For instance, at position 1659, most G1 and G2 sequences carry a T instead of the C used in the primer, and at position 1662, G4 sequences often carry an A instead of the G present in the primer. These mismatches can significantly reduce the amplification efficiency of non-targeted genotypes. In this context, including degenerate bases at key positions—particularly at 1359 in F1 and 1659 in R1—would have mitigated some of the observed bias.

Altogether, the reliance on a single, non-degenerate primer pair targeting a polymorphic region may have led to an important amplification bias, resulting in underrepresentation or complete absence of certain HCV subtypes. This limitation is particularly critical given that the study relies on PCR products to generate consensus sequences for phylogenetic analyses. Thus, the results and conclusions regarding circulating subtypes and transmission clusters should be interpreted with caution, as they may reflect primer-driven selection rather than the true underlying viral diversity.

Statistical Analyses: Using outcome information to select features (supervised selection) can lead to overfitting and overly optimistic estimates of model performance, especially if the same dataset is used for both feature selection and model fitting, as this process may capitalize on random associations present in the training data rather than true underlying relationships, resulting in poor generalization to new data. To address these issues, regularization techniques such as LASSO or ridge regression offer more robust approaches to variable selection and help manage multicollinearity, while the use of cross-validation or bootstrap methods during variable selection and model evaluation can further reduce the risk of overfitting and provide more realistic estimates of model performance. Furthermore, the cutoff (e.g., $p < 0.10$) is arbitrary and may exclude relevant variables or include irrelevant ones, depending on sample size and power.

The statistical methods should be improved to limit overfitting as much as possible.

Lines 317-318 “a rare case of infection was found with a major population of genotype 3a and minor populations of genotypes 1a and 2b.” Do the authors have access to patients' history, including details on how the infection might have occurred?

Lines 456-460: “The degree of clustering observed within this study at 29.5% is similar to that observed from other injecting drug cohorts. For example, a study examining HCV transmission across four Indian cities revealed that 28.8% of HCV sequences clustered, 49 while studies conducted in North America have revealed variable levels of clustering from 46% in Baltimore⁵⁰, 33% in Wisconsin¹⁸, 25% in New York⁵¹, 31% in Vancouver⁵² and 36% in Ottawa⁵³. “

Differences in clustering rates across studies may partly reflect the use of different primer sets and target regions. Notably, studies reporting higher clustering rates, such as those from Baltimore and Ottawa, often used more conserved genomic regions, whereas studies using hypervariable regions (e.g., HVR1) typically observe lower clustering due to greater sequence diversity

Minor changes:

Line 69: (PWUD)/, should be a point: (PWUD).

Version 1:

Reviewer comments:

Reviewer #1

(Remarks to the Author)

I am satisfied with the response to my comments - thank you for taking the time to revise the manuscript.

There is a minor typo the unadjusted OR for Partner, spouse, boyfriend, girlfriend in Table 3 - 2.069 (1169., 3.663). I think the decimal point is in the wrong place.

Reviewer #2

(Remarks to the Author)

Thank you for successfully addressing all my concerns.

REVIEWER COMMENTS

Reviewer #1 (Remarks to the Author):

Whilst I appreciate that the authors used a high-fidelity polymerase for their PCR reactions, I would be slightly concerned about the error rate in the sequencing after putting the samples through 48 cycles of PCR plus Illumina sequencing which also requires additional amplification. Was there any data to indicate this was not an issue?

We appreciate the reviewer's comment regarding potential error accumulation through multiple rounds of PCR and Illumina sequencing. While multiple PCR cycles can, in principle, introduce errors, the use of a high-fidelity polymerase (error rate $\sim 10^{-5}$ – 10^{-6} per base per cycle) ensures that PCR-induced substitutions occur at a very low frequency relative to natural HCV diversity. Early-cycle errors, if introduced, typically manifest only as rare singletons or low-frequency haplotypes, which are removed by GHOST's clustering and entropy-based filters and therefore do not influence transmission linkage detection. For our 306-bp amplicon, the expected cumulative error rate from reverse transcription and PCR amplification corresponds to approximately 0.15–0.31 substitutions per amplicon on average. In practice, this means that most artifactual changes occur at very low frequency. For two independent samples to be incorrectly linked, both would need to acquire the same spurious substitution (or a closely related change) at sufficient abundance for the resulting erroneous haplotype to dominate the minimal genetic distance calculation. Given the random nature of RT and PCR misincorporations, combined with GHOST's clustering, filtering, and error-correction procedures, the probability of this occurring is exceedingly low.

The GHOST assay is designed to detect transmission linkages using all intra-host HCV variants identified by NGS. Although sequencing errors may generate artificial variants, they cannot create false linkages between distantly related strains or separate closely related ones, as genetic relatedness is assessed by minimal distances across the complete set of variants. This robustness has been validated in computational simulations with artificially introduced errors, which demonstrated no impact on detection accuracy. In addition, the GHOST pipeline incorporates a dedicated error-correction module, including the CASPER algorithm (Kwon et al., *BMC Bioinformatics* 2014), which corrects substitution errors using a context-aware k-mer approach. In our implementation, CASPER was run with a quality threshold of 15, k-mer length of 17, k-mer neighborhood of 8, and a minimum match threshold of 95%, as described by Longmire et al. (*BMC Genomics* 2017). Together with the use of a high-fidelity polymerase, these steps ensure that sequencing artifacts do not compromise the accuracy of transmission linkage detection.

It would be useful to provide the size of the HVR1 amplicon generated with the PCR as it is helpful to understand for several of the analytical steps.

The size of the amplicon is 306 bp and this is now mentioned in the revised manuscript.

An 18% failure rate for a PCR is very high if samples are known to be RNA positive – was the PCR used in this study well validated in previous work? Highly sensitive PCRs can be developed for the E1/E2 region, so this failure rate is concerning.

We would like to clarify that the 18% figure reported in the manuscript refers to the proportion of samples that either failed PCR or did not produce sufficient sequence to pass GHOST QC for inclusion. The PCR-only failure rate was 11.4% overall (133/1,201 samples), with site-specific rates ranging from 3.37% (Wisconsin) to 46.15% (Illinois)

(Supplementary Table 1). Variation between sites is largely attributable to pre-analytical factors affecting RNA integrity, including delayed centrifugation/freezing or prolonged room-temperature storage, as documented in site logs. The Allen group at the Ragon Institute of MGH, MIT and Harvard has been sequencing HCV for 3 decades using a wide variety of amplification and sequencing strategies, and the PCR method used here has been extensively validated in prior work and by the CDC in extensive published works. In a retrospective study from Wisconsin (2016–2017), the PCR failure rate was only 5%, and from samples with known viral loads we have successfully amplified and sequenced specimens with <100 IU/mL (see Hochstatter KR, Tully DC et al. *Emerging Infectious Diseases* 2021). Thus, in previous applications, this approach has not yielded significant differences in amplification success, further supporting that the higher failure rates observed in certain sites are due to sample handling and integrity rather than inherent assay limitations.

For clarity we have now revised the manuscript to state: “From 1,201 specimens, 216 (18.0%) either failed PCR or did not generate sufficient sequence to pass GHOST QC for inclusion. The PCR-only failure rate was 11.4% overall, varying by site (3.37–46.15%; Supplementary Table 1), largely reflecting pre-analytical differences in specimen handling and RNA integrity.”

Data availability – are the authors planning to submit consensus sequences to Genbank or another appropriate data repository? The reasons for not doing this are unclear to me, and sequences can be held under embargo prior to publication of the manuscript if required.

We thank the reviewer for this important suggestion regarding sequence data deposition. Our initial hesitation stemmed from concerns around participant consent and whether our IRB would permit deposition of consensus sequences. However, we fully agree that public data availability is essential for transparency and reproducibility. We have now further anonymized the data in accordance with IRB guidance and deposited all consensus sequences in GenBank under accession numbers PX261275 - PX261966 with an embargoed release date. The accession numbers have been added to the Data Availability section.

Figure 1 – unclear where the genotype distribution data is in the figure?

Reference to genotype distribution has now been removed as this was from an older version of this figure legend, so it now reads: “Sampling overview of HCV from study sites across the United States.”

The authors state that – “Almost one-fifth (19%) cleared HCV infection with treatment.”. Where were they collecting this data from, were participants followed up at a later date? Does this mean four-fifths did not receive treatment or failed treatment?

This information was obtained from harmonized questionnaires administered to all participants across study locations. Specifically, participants were asked, “Have you ever been told that your hepatitis C had cleared or that you have been cured?” There was no follow-up or clinical confirmation of these responses. The finding does not imply that four-fifths of participants failed treatment or did not receive it — response options included Yes, No, Not Filled Out, Not Asked/Removed, Don’t Know, or Refused. As shown in Table 2, 43 participants answered “Yes” to this question, while 175 answered “No.” In addition, as previously reported in another study using the same cohort that only 12% of participants ever received anti-HCV medication (Estadt, A.T., Kline, D., Miller, W.C. *et al.* Differences in hepatitis C virus (HCV) testing and treatment by opioid, stimulant, and polysubstance use among people who use drugs in rural U.S. communities. *Harm Reduct J* **21**, 214 (2024)). As discussed in the manuscript there are multilevel social factors that may be responsible for the extremely limited rural access to HCV treatment for PWUD including a lack of HCV specialty providers such as hepatologists, gastroenterologists, and infectious disease physicians who are more likely to be in urban areas (e.g. Rongey C, et al. Impact of rural residence and health system structure on quality of liver care. *PLoS*

ONE. 2013). Als, there are certain national policies which may be a barrier for HCV preventive and treatment services such patients were required to adhere to sobriety requirements to receive HCV treatment, especially those insured by state Medicaid programs which covered the majority (64%) of our study participants (Campbell CA, et al. State HCV incidence and policies related to HCV Preventive and Treatment Services for Persons Who Inject Drugs - United States, 2015–2016. MMWR Morb Mortal Wkly Rep. 2017).

Figure 2 – did the authors find any evidence of mixed infections of the same genotype or was the approach used only able to identify mixed genotype/subtype infections? I would expect these to also be present among active PWUD?

We thank the reviewer for raising this important point. While our pipeline is capable of identifying mixed infections at the genotype or subtype level, detecting intra-genotype mixtures particularly those involving closely related lineages is notably challenging using amplicon-based consensus sequencing. Within genotype mixed infections are more challenging to resolve due to the rate of evolution and the R1/E2 region's small size. Although we did find limited evidence of a 1a/1b mixed infection some samples in our dataset did show elevated intra-host diversity suggestive of possible within-genotype mixtures but these could not be robustly separated from normal within-host variation and there was limited phylogenetic evidence over the E1/E2 amplicon for any such events. In fact, reports of true mixed infections within the same genotype exist but are relatively rare from an examination of published studies.

Nevertheless, we have added the following statement in the discussion to reflect this point: “While the GHOST pipeline can identify mixed infections it should be noted that the amplicon size and the high rate of evolution may limit resolution in distinguishing true mixed infections within the same genotype without complementary deep sequencing of another genomic region.”

Figure 3 – Presentation of this tree could be improved, as the gt1b clade appears to be in the middle of the gt1a sequences? Based on the number of sequences, I assume the sequences below the gt1b cluster are gt1a but this is not clear and should be more clearly indicated. Again amplicon length would be helpful here and some evidence of bootstrapping. How were consensus sequences derived for mixed gt infections?

We thank the reviewer for these suggestions regarding Figure 3. We have now revised the phylogenetic tree to improve clarity by clearing annotating the genotypes as opposed to the sampling site because the goal of this figure was to highlight the genotypic diversity found as opposed to how consensus sequences clustered.

Regarding consensus sequence derivation for mixed-genotype infections: in cases where mixed-genotype infections were detected (based on genotyping analyses), we generated separate consensus sequences for each distinct genotype using our analysis pipeline, which resolves mixed infections by clustering reads into genotype-specific assemblies. Only consensus sequences with $\geq 90\%$ genome coverage and high read depth per position were retained for further analysis.

The amplicon length is 306 nucleotides and this has now been added to the figure legend and bootstrapping values have been added to the figure.

GHOST analysis – the authors should provide some more details here. Were these samples all one particular genotype or are multiple genotypes shown in the figure? Presumably linkage between different genotypes is not possible, so this is helpful to understand. More information on what the single dots would be helpful – are these unlinked samples? Also the authors used an empirically defined threshold value of 0.037 to define linkage – did the authors consider doing any sensitivity analysis on this (ie if this is adjusted slightly do you see more linkage?)

We apologize to the reviewer for the lack of clarity here. The reviewer is indeed correct that all samples are the major genotype and multiple genotypes are not shown in this figure as linkage between different genotypes is naturally not possible. We now also explain in the figure legend that the single unconnected dots represent samples for which no genetic linkage was detected within the specified threshold.

Regarding the threshold value of 0.037, this is the empirically derived pairwise genetic distance cut-off validated in prior CDC work (Campo et al., 2016; Oster et al., 2018) and has been widely applied in HCV transmission network analyses. This threshold is based on a wealth of combined epidemiological and molecular data confirming or rejecting relatedness, accumulated over 2 decades. This transmission linkage cutoff is the pre-set value in the GHOST informatics pipeline and includes the standard deviation of the distances. Increased threshold results in reduction of detection accuracy for HCV cases linked by transmission in HCV outbreaks. However, GHOST has a model, Surveillance Assistance Report, which uses an additional threshold (0.05), but this model is not applicable to outbreaks and intended to be used for the detection of high-risk populations. However, if we extend the threshold to 0.05 then we increase the number of clusters by 14.

Study Site	Additional number of clusters at 0.05 threshold
WV	0
OR	1
WI	3
OH	0
NC	2
KY	6
New England	2

I think the analysis on viral diversity is confounded by the use of amplicons for deep sequencing – as the method to obtain the sequences can introduce errors into the sequences, so the authors need to be more nuanced in their conclusions in this section and discuss this as a limitation.

While amplicon-based sequencing provides high sensitivity for viral genome recovery, it can introduce biases that confound estimates of intrahost diversity. Primer-template mismatches and uneven amplification efficiencies may lead to underrepresentation of certain variants, while PCR and sequencing errors can artificially inflate diversity estimates. In addition, the use of short amplicons limits the ability to infer linkage between distant mutations. We have now added this as a limitation in the discussion and created a more nuanced conclusion in the results section to reflect this point.

“Taken together, this suggests that those participants not found to be in a transmission cluster may have an infection reminiscent of a longer timeframe (i.e. more chronic like stage of infection) compared to those within clusters who appear to be harboring less diversity which is a known attribute of the early stages of infection. However, this interpretation should be treated with caution: viral diversity can also be influenced by host immune pressure, treatment history, viral genotype, and potential technical biases introduced by amplicon sequencing. The observed site-specific differences in intra-host viral diversity likely reflect a combination of underlying epidemiological dynamics, transmission patterns and sampling variation rather than infection stage along.”

Table 1 – I would encourage the use of some statistics here to demonstrate the similarity between the subsets of the cohorts. The raw data is insufficient to illustrate this?

We thank the reviewer for this helpful suggestion. In the revised manuscript, we have updated Table 1 to include statistical comparisons between (i) participants who provided samples vs those who did not, and (ii) participants with sequenced samples, participants with non-sequenced samples, and those without samples. These analyses show that, while some differences exist (notably for gender, race/ethnicity, homelessness, and geography), the sequenced cohort is broadly comparable to the overall enrolled population. We have the method section accordingly and added text in the Results to acknowledge these differences and to note that they are consistent with expected epidemiological patterns in PWUD populations and do not undermine the validity of our phylogenetic inferences.

Reviewer #2 (Remarks to the Author):

This is a cross-sectional study conducted among people who used drugs (PWUD) in rural counties, recruited between 2018 and 2021. The study includes individuals residing in the study area who reported any past 30-day injection drug use and/or non-injecting opioid use “to get high” (including heroin and prescription pain medications). Participants were recruited using respondent-driven sampling to characterize the circulating HCV strains circulating among PWUD in rural US areas and to identify factors associated with HCV transmission clusters. The results include a description of the prevalence of HCV subtypes in the rural PWUD population and their phylogenetic clustering, having a very interesting discussion.

We thank the reviewer for their careful summary of our study and for highlighting the value of our discussion.

Comments:

While the study addresses an important public health issue, it lacks elements of novelty that would significantly enhance its contribution to the field. To bring added value to this type of molecular epidemiology work, it would have been highly informative to include detailed epidemiological data linking participants, such as information on shared injection practices, paraphernalia, or other risk behaviors. Establishing clearer epidemiological relationships between participants and the identified transmission clusters is crucial to strengthen the interpretation of the phylogenetic findings.

We thank the reviewer for this insightful comment. We agree that detailed epidemiological data such as shared injection practices or paraphernalia use may strengthen the linkage between participants and phylogenetic clusters. However, such data were not consistently collected across ROI sites due to differences in local IRB requirements, participant sensitivity around risk behaviors, and the need to harmonize questionnaires across eight states. Despite this limitation, our study represents one of the largest rural PWUD cohorts analyzed with molecular epidemiology methods to date and demonstrates clear patterns of HCV clustering across geographically distinct regions. We believe these findings add important novel insights into rural HCV transmission dynamics and provide a foundation for future studies incorporating more granular behavioral data. In fact, this is the first such report on the molecular epidemiology of HCV at a multi-state level within the United States during a concurrent opioid crisis.

In addition, our prior work (Tully DC et al., *Clinical Infectious Diseases*, 2022) highlights that even when detailed sexual and injecting partnership data are available, phylogenetic linkage is not always observed. Specifically, 63% of PWUDs did not share a genetically related infection with their reported partner, despite extensive structured interviews. This underscores that epidemiological relationships do not always map directly onto phylogenetic clustering. Our previous studies, and those of others (e.g. Rose R. et al., *Infection, Genetics and Evolution*, 2019), further demonstrate rapid cycles of clearance and reinfection, or undetected chronic infections, can complicate epidemiological–genetic concordance. These complexities emphasize the value of molecular approaches such as GHOST for uncovering hidden transmission network dynamics and for forming the framework for more efficient targeted interventions of hepatitis c transmission as demonstrated by Campo DS and Khudyakov Y (*Infect Genet Evol.* 2018).

In addition, given the inherent intra-host variability of HCV, phylogenetic analysis using consensus sequences may not adequately capture all transmission relationships particularly when attempting to define clusters among PWUD-related HCV transmissions. This limitation is especially relevant due to the potential bottlenecking transmissions, where a subpopulation of the viral quasispecies is passed from one individual to another.

We thank the reviewer for raising this important point. We agree that consensus-only approaches can obscure intra-host variation and underestimate transmission relationships, particularly in the context of bottlenecked PWUD-related transmissions. However, we would like to clarify that the GHOST pipeline does not rely on consensus sequences. Instead, it incorporates quasispecies-level data, capturing within-host diversity to assess genetic relatedness between samples. This approach has been validated extensively by the CDC in outbreak investigations and provides a higher resolution for detecting transmission links than consensus phylogenies alone. In our analysis, consensus trees were used primarily for subtype assignment and visualization of broad phylogenetic clustering, while GHOST quasispecies analysis formed the basis for identifying putative transmission clusters. We have revised the Discussion to make this clearer and to explicitly acknowledge that, while GHOST enhances sensitivity over consensus-only methods, very fine-scale transmission dynamics may still benefit from complementary approaches such as full-genome haplotype reconstruction. To our knowledge, this is among the first studies to leverage intra-host sequence variation, rather than consensus sequences alone, for cluster identification in HCV.

Furthermore, analyzing multiple genomic regions with differing levels of variability (e.g., a highly variable region such as E1/E2 or HVR1, an intermediate like NS5B, and a more conserved region like core) would have improved the resolution and robustness of the phylogenetic inferences.

We thank the reviewer for this insightful suggestion. We agree that sequencing and analysing multiple genomic regions with different evolutionary rates (e.g., E1/E2 or HVR1, NS5B, and Core) can improve the robustness and resolution of phylogenetic inferences. However, the scope of this study was to leverage the GHOST platform, which has been optimized and validated by CDC for outbreak investigations using a standardized E1/E2 fragment. This region was selected because of its high variability and proven ability to resolve transmission clusters in epidemiological contexts. We acknowledge that this approach does not capture the full spectrum of HCV genomic diversity, and we have added this point as a limitation in the discussion. Future work, particularly with next-generation sequencing approaches covering multiple regions or near full-length genomes, will be important to validate and extend these findings, especially in relation to finer-scale phylogenetic resolution and antiviral resistance monitoring.

All these samples have been shipped to the CDC for long-term storage and due to the current U.S administration placing the Division of Viral Hepatitis staff on administrative leave the current fate of these samples remains uncertain. As a result, it is unfortunately not possible to re-amplify or sequence additional genomic regions at this time.

If the transmission event occurred recently, phylogenetic analysis using the highest variable region has strong discriminatory power to infer relatedness between samples, generating a clear transmission cluster. However, if the transmission occurred several months or years ago, the quasispecies in each host may have diverged significantly, requiring the study of a viral genomic region with median or lower variability to detect phylogenetic links.

The reviewer is correct that the transmission events that we determined are of recent transmission events which we believe may be relevant for targeted public health interventions. Deep sequencing of HCV for the identification of transmission clusters is not regularly performed with most studies using consensus sequences to derive transmission clusters. We believe that using the entire viral population within a sample is more sensitive to the detection of both major and minor transmission events. GHOST builds k-step networks from all the unique haplotypes found within a

participants and allows us to visualise the structure of intra-host HCV populations. The k-step network provides a quick and intuitive visualization of all haplotypes detected at a frequency equal to or greater than the frequency minimum (10 or $\geq 1\%$, whichever is greater) in either sample, and genetic linkage among these haplotypes. Links are drawn only for the genetic distances below the distance threshold (0.037) used by GHOST to infer transmission events. Links are shown both within and between sampled intra-host HCV populations.

In this example on the left network haplotypes in red indicate shared haplotypes by both samples while green represents unique haplotypes from the 2nd sample that are not shared with the first sample while blue represents unique haplotypes from the first sample that are not shared with the second sample. With every putative the strength of the link is assessed by the frequency-weighted percentage of inter-host pairwise sequence comparisons

having a distance below the GHOST related threshold. In this example, the link strength is strong with 74% of haplotypes shared between participants.

We have modified throughout the manuscript that our transmission clusters should be interpreted primarily as evidence of ongoing or recent transmission events rather than reflecting longer-term or historical transmission events.

While the authors correctly target a relatively conserved region within the E1/E2 region of the HCV genome, their use of a single primer pair, without degenerate positions, raises concerns about potential amplification bias across subtypes. Although this region is overall conserved, several subtype-specific nucleotide polymorphisms are well documented and can significantly impact primer annealing efficiency.

In the forward primer (F1: 5'-GGATATGATGATGAACTGGT-3'; positions 1356–1375), position 1359 is often a C in most genotype 1 (G1) sequences, whereas the primer sequence includes a T, potentially reducing amplification efficiency in G1 viruses. Additionally, genotypes 2b and 3 commonly present two or more mismatches within this region, and genotype 4 strains often carry a mismatch at the 3'-end (position 1375), where they consistently have an A, further impairing amplification. Notably, the primer may favor amplification of G3 viruses, which typically retain a T at position 1359, resulting in overrepresentation of this subtype in downstream analyses.

The reverse primer (R1: 5'-ATGTGCCAGCTGCCGTTGGTGT-3'; positions 1652–1673), although designed in a relatively conserved region as well, also contains subtype-specific mismatches. For instance, at position 1659, most G1 and G2 sequences carry a T instead of the C used in the primer, and at position 1662, G4 sequences often carry an A instead of the G present in the primer. These mismatches can significantly reduce the amplification efficiency of non-targeted genotypes. In this context, including degenerate bases at key positions—particularly at 1359 in F1 and 1659 in R1—would have mitigated some of the observed bias.

Altogether, the reliance on a single, non-degenerate primer pair targeting a polymorphic region may have led to an important amplification bias, resulting in underrepresentation or complete absence of certain HCV subtypes. This limitation is particularly critical given that the study relies on PCR products to generate consensus sequences for phylogenetic analyses. Thus, the results and conclusions regarding circulating subtypes and transmission clusters should be interpreted with caution, as they may reflect primer-driven selection rather than the true underlying viral diversity.

The reviewer provides a technically accurate description of potential genotype biases that can arise from primer structure. Primer mismatches can indeed weaken amplification efficiency for certain viral sequence variants. However, such effects are often mitigated by factors such as nucleotide stacking with adjacent G/C bases, specific PCR conditions, and the type of application (e.g., PCR detection vs. sequencing). More importantly, the practical performance of the experimental protocol must be considered.

This particular assay has a long record of use in the field. These primers have been employed by the CDC for nearly 30 years in outbreak investigations and were extensively tested and validated across HCV genotypes 1–6, with no significant biases observed. This broad validation explains their longstanding use to test specimens from the U.S. and multiple countries in South America, Africa, and Asia.

To further minimize potential biases, including those noted by the reviewer, studies have investigated the impact of primer concentration, annealing temperature, and cycling conditions—factors that strongly influence melting temperature and PCR efficiency. In addition, the assay uses nested PCR, which helps reduce the impact of mismatches.

Finally, it is important to clarify that the goal of the GHOST assay is not precise genotyping, but rather the detection of genetic linkages among HCV strains or the identification of genetically related strains by sequencing. These are typically encountered in outbreak investigations and are associated with direct HCV transmission. The GHOST protocol relies on next-generation sequencing and the comparison of all detected intra-host sequence variants, rather than on consensus sequences as suggested by the reviewer.

Statistical Analyses: Using outcome information to select features (supervised selection) can lead to overfitting and overly optimistic estimates of model performance, especially if the same dataset is used for both feature selection and model fitting, as this process may capitalize on random associations present in the training data rather than true underlying relationships, resulting in poor generalization to new data. To address these issues, regularization techniques such as LASSO or ridge regression offer more robust approaches to variable selection and help manage multicollinearity, while the use of cross-validation or bootstrap methods during variable selection and model evaluation can further reduce the risk of overfitting and provide more realistic estimates of model performance. Furthermore, the cutoff (e.g., $p < 0.10$) is arbitrary and may exclude relevant variables or include irrelevant ones, depending on sample size and power. The statistical methods should be improved to limit overfitting as much as possible.

We agree with the reviewer that conventional outcome-based variable selection risks overfitting. In our initial submission we presented standard logistic regression results (Table 3), but in response to this concern we have now re-analyzed the data using LASSO penalized logistic regression with 10-fold cross-validation for penalty tuning, complemented by bootstrap stability analysis ($B=200$). This approach simultaneously performs variable selection and shrinkage, reducing overfitting and providing more conservative estimates. The penalized model achieved a mean cross-validated ROC-AUC of 0.60 (SD 0.08), confirming only modest discrimination for cluster membership. After shrinkage, most covariates were reduced to zero. A small subset of predictors (e.g., younger age, methamphetamine vs opioids, recruitment through partner/spouse or peers, illegal income, and syringe source from drug dealers) were

retained, but effect sizes were modest and bootstrap selection frequencies ranged from ~55–86%. These results are now presented in Supplementary Table 2 which combine odds ratios and 95% CIs for clarity. We have retained Table 3 showing the conventional logistic regression in the main text for reasons of transparency and comparability. Most studies in this field present univariable and multivariable logistic regression results and retaining Table 3 allows readers to directly compare our findings with prior literature.

We have revised the Abstract, Methods, Results, and Discussion accordingly to highlight that while some individual-level factors appear modestly associated with clustering in conventional models, penalized regression demonstrates limited predictive signal, suggesting that clustering is more strongly shaped by social and structural contexts than by individual characteristics.

Lines 317-318 “a rare case of infection was found with a major population of genotype 3a and minor populations of genotypes 1a and 2b.” Do the authors have access to patients’ history, including details on how the infection might have occurred?

We appreciate the reviewer’s comment. Unfortunately, detailed clinical or behavioral data (e.g., patient history, injection practices, or potential exposure routes) were not available for this particular case, as these data were not collected in a manner that could be linked back to individual sequences due to confidentiality and study design restrictions. Therefore, we are unable to comment directly on how this mixed-genotype infection may have occurred.

Lines 456-460:”The degree of clustering observed within this study at 29.5% is similar to that observed from other injecting drug cohorts. For example, a study examining HCV transmission across four Indian cities revealed that 28.8% of HCV sequences clustered, 49 while studies conducted in North America have revealed variable levels of clustering from 46% in Baltimore⁵⁰ , 33% in Wisconsin¹⁸, 25% in New York⁵¹, 31% in Vancouver⁵² and 36% in Ottawa⁵³. “

Differences in clustering rates across studies may partly reflect the use of different primer sets and target regions. Notably, studies reporting higher clustering rates, such as those from Baltimore and Ottawa, often used more conserved genomic regions, whereas studies using hypervariable regions (e.g., HVR1) typically observe lower clustering due to greater sequence diversity

In our discussion we have stated that “The elevated clustering rate observed in these known partnerships may be attributed to the study designs, which involves more frequent assessments, thereby increasing the likelihood of capturing transmission events early. Differences in the rate of clustering among participants could reflect regional differences in drug use networks, recruitment approaches, sampling density, clustering analyses methods or behavioral differences. ” and we have now added an additional sentence reflecting the reviewers concern: “. It may also be attributed to the use of different amplification strategies involving different primer sets and target regions.”

Minor changes:

Line 69: (PWUD)/, should be a point: (PWUD).

This has now been modified in the revised manuscript.

REVIEWERS' COMMENTS

Reviewer #1 (Remarks to the Author):

I am satisfied with the response to my comments - thank you for taking the time to revise the manuscript.

There is a minor typo the unadjusted OR for Partner, spouse, boyfriend, girlfriend in Table 3 - 2.069 (1169., 3.663). I think the decimal point is in the wrong place.

We thank the reviewer for pointing out this typo. This has now been modified in a revised version of the manuscript.

Reviewer #2 (Remarks to the Author):

Thank you for successfully addressing all my concerns.